 

# Fluctuations of the transcription factor ATML1 generate the pattern of giant cells in the *Arabidopsis* sepal

Heather M Meyer[1,2†], José Teles[3†], Pau Formosa-Jordan[3†], Yassin Refahi[3], Rita San-Bento[4], Gwyneth Ingram[4], Henrik Jönsson[3,5,6*], James C W Locke[3,7,8*], Adrienne H K Roeder[1,2,9*]

[1]Weill Institute for Cell and Molecular Biology, Cornell University, United States; [2]The graduate field of Genetics, Genomics, and Development, Cornell University, Ithaca, United States; [3]Sainsbury Laboratory, University of Cambridge, Cambridge, United Kingdom; [4]Laboratoire Reproduction et Développement des Plantes, Univ Lyon, ENS de Lyon, UCB Lyon 1, CNRS, INRA, Lyon, France; [5]Computational Biology and Biological Physics, Lund University, Lund, Sweden; [6]Department of Applied Mathematics and Theoretical Physics, University of Cambridge, Cambridge, United Kingdom; [7]Department of Biochemistry, University of Cambridge, Cambridge, United Kingdom; [8]Microsoft Research, Cambridge, United Kingdom; [9]Section of Plant Biology, School of Integrative Plant Science, Cornell University, Ithaca, United States

**\*For correspondence:** Henrik. Jonsson@slcu.cam.ac.uk (HJ); James.Locke@slcu.cam.ac.uk (JCWL); ahr75@cornell.edu (AHKR)

[†]These authors contributed equally to this work

**Competing interests:** The authors declare that no competing interests exist.

**Abstract** Multicellular development produces patterns of specialized cell types. Yet, it is often unclear how individual cells within a field of identical cells initiate the patterning process. Using live imaging, quantitative image analyses and modeling, we show that during *Arabidopsis thaliana* sepal development, fluctuations in the concentration of the transcription factor ATML1 pattern a field of identical epidermal cells to differentiate into giant cells interspersed between smaller cells. We find that ATML1 is expressed in all epidermal cells. However, its level fluctuates in each of these cells. If ATML1 levels surpass a threshold during the G2 phase of the cell cycle, the cell will likely enter a state of endoreduplication and become giant. Otherwise, the cell divides. Our results demonstrate a fluctuation-driven patterning mechanism for how cell fate decisions can be initiated through a random yet tightly regulated process.

## Introduction

One of the fundamental questions in developmental biology is how patterns of specialized cell types are formed *de novo* from a field of identical cells. Wolpert's French flag model proposes that a group of identical cells differentiate into different cell types based on threshold concentrations of a morphogen gradient (*Wolpert, 1996*). Each cell responds to the morphogen individually by expressing specific sets of downstream genes determined by the concentration sensed. This model has successfully explained the formation of various animal tissue patterns ranging from Bicoid anterior-posterior patterning in *Drosophila* to BMP dorsal-ventral axis patterning in *Xenopus* (*Eldar et al., 2002*; *Houchmandzadeh et al., 2002*; *Kondo and Miura, 2010*; *Spirov et al., 2009*; *Tucker et al., 2008*). In plants, traditional morphogens have yet to be observed, although it has been argued that the phytohormone auxin acts as an atypical morphogen that is actively transported to regulate plant morphogenesis (*Bhalerao and Bennett, 2003*).

**eLife digest** Plant and animal organs contain several types of cells that perform different roles. As a plant or animal develops, these different cell types can form intricate patterns. To start the pattern, a few cells within a group of identical cells must somehow become different from their neighbors. Some patterns of cells are very well organized and easily reproduced. However, sometimes cells spontaneously become different from their neighbors, producing a less ordered pattern.

In a plant called *Arabidopsis* (commonly known as Thale cress), a scattered pattern of giant cells and small cells spontaneously forms within a part of the developing flower called the sepal. A protein called ATML1 is a key regulator in the formation of giant cells, but because it is found in both giant cells and small cells, it is not clear how this regulation works.

Mathematical models of this process suggest that identical cells could initially acquire subtle differences, potentially from random fluctuations in the activity of key regulatory molecules, to start the patterning process. Meyer, Teles, Formosa-Jordan et al. used a combination of microscopy, image analysis and mathematical modeling to investigate how the level of ATML1 fluctuates in cells to give rise to the pattern within the sepal. The experiments show that early in the development of the sepal, the levels of ATML1 fluctuate up and down in every sepal cell. If ATML1 reaches a high level specifically when a cell is preparing to divide, that cell will decide to become a giant cell, whereas if the level of ATML1 is low at this point, then the cell will divide and remain small.

Overall, the findings of Meyer, Teles, Formosa-Jordan et al. demonstrate that fluctuations of key regulators while cells are preparing to divide are important for creating patterns during development. A future challenge is to examine whether other tissues in plants, or tissues in other organisms, use a similar mechanism to generate patterns of cells.

In contrast to the morphogen gradient paradigm, many patterning phenomena seem to lack specific localized signaling cues. In these cases, it is not known how identical cells become slightly different from their neighbors to initiate the patterning process. Theoretical approaches suggest a role for small differences of key transcriptional regulators, generated for example by stochastic fluctuations (*Collier et al., 1996*; *Hülskamp and Schnittger, 1998*; *Hülskamp, 2004*; *Meinhardt and Gierer, 1974*; *Turing, 1952*). In these models, subtle initial differences between identical neighboring cells in activators and inhibitors are amplified and solidified through regulatory feedback loops and cell-to-cell communication to establish different cell fates (*Kondo and Miura, 2010*; *Meyer and Roeder, 2014*). For instance, in a computational model of lateral inhibition where Notch and Delta mutually inhibit one another in the same cell, small stochastic changes in Notch or Delta can flip a switch between cell identities (*Sprinzak et al., 2010*). Subtle concentration changes in Notch or Delta may change a cell's signaling ability and either push cells into a sending state (i.e. high Delta/ low Notch) or a receiving state (i.e. high Notch/low Delta). These changes subsequently are amplified through cell-to-cell Notch-Delta signaling to create ordered patterns (*Collier et al., 1996*; *Formosa-Jordan and Ibañes, 2014*; *Sprinzak et al., 2010*). While manipulating Notch-Delta levels in individual mammalian cells supports this model (*Matsuda et al., 2015*; *Sprinzak et al., 2010*), these dynamic fluctuations are difficult to detect during tissue patterning within a multicellular system. A similar lateral inhibition model has been proposed to explain trichome (i.e. hair cell) spacing in plants (*Digiuni et al., 2008*; *Hülskamp and Schnittger, 1998*; *Hülskamp, 2004*; *Meinhardt and Gierer, 1974*). In these trichome models, initially identical cells can acquire subtle differences through brief stochastic fluctuations of transcriptional activators. These activators amplify both their own expression and the expression of faster-diffusing transcriptional repressors that move to the neighboring cell to create a non-random distribution of trichomes, following a Turing-like model (*Hülskamp, 2004*; *Meinhardt and Gierer, 1974*; *Turing, 1952*). Several transcriptional regulators needed for trichome patterning have been identified that support this model (*Bouyer et al., 2008*; *Greese et al., 2014*; *Hülskamp and Schnittger, 1998*; *Hülskamp, 2004*; *Schellmann et al., 2002*). However, the stochastic fluctuations of these genes remain to be observed *in vivo* during trichome development.

Most biological examples of stochasticity focus on how noise is buffered during development, suggesting that multiple species have evolved genetic regulatory mechanisms to offset the potentially detrimental effects of noisy gene expression (*Abley et al., 2016*; *Arias and Hayward, 2006*; *Besnard et al., 2014*; *Heisler et al., 2005*; *Houchmandzadeh et al., 2002*; *Howell et al., 2012*; *Jönsson et al., 2006*; *Meyer and Roeder, 2014*; *Raj et al., 2010*; *Reinhardt et al., 2003*; *Smith et al., 2006*). However, a few studies have demonstrated the importance of stochasticity in creating the correct distribution of phenotypes within a population of cells. For instance, during *Drosophila* retinal development, the transcriptional regulator *spineless* stochastically turns on or off to generate a proportional but randomly distributed population of photoreceptor subtypes (~30% ultraviolet/blue sensitive and ~70% ultraviolet/green sensitive; *Wernet et al., 2006*). Without the stochastic dynamics of *spineless* expression, all cells adopt the same fate (*Wernet et al., 2006*; *Johnston and Desplan, 2014*). Similarly, a stochastic Markov model illustrates how a tumor can maintain phenotypic equilibrium between different cancer cell subpopulations. In this model, isolated cancer subpopulations will return to their respective proportions over time through stochastic interconversions (*Gupta et al., 2011*). These studies suggest that stochasticity can help different cell populations to reach or maintain the correct phenotypic equilibrium.

During the development of *Arabidopsis thaliana*'s outmost floral organ, the sepal, equivalent epidermal cells in the primordium differentiate to produce a scattered pattern of giant cells that are interspersed between smaller cells (*Figure 1A–F*; *Roeder et al., 2010*, *2012*; *Tauriello et al., 2015*). The sepal is a useful model system because the giant cell patterning process can be live imaged from the earliest stages of initiation through giant cell differentiation. At maturity, giant cells are approximately one-fifth the length of the sepal and form when an epidermal cell undergoes multiple rounds of endoreduplication, an alternative cell cycle in which a cell replicates its DNA without undergoing mitotic division (*Figure 1C–G*; *Roeder et al., 2010*). Mature sepals typically contain the same proportion of giant cells relative to small cells, although their spatial distribution varies from sepal to sepal and giant cells may even form adjacent to one another (*Figure 1C–F*). The correct proportion of giant cells and small cells is needed to control the curvature of the sepal; when the proportion of giant cells is altered, sepals are unable to enclose and protect the developing floral organs (*Roeder et al., 2010*, *2012*). Thus, we ask how giant cell patterning initiates and reproducibly produces the correct proportion of giant cells for proper sepal curvature?

We have previously shown that giant cells do not form on the sepal epidermis in plants with loss-of-function mutations in *Arabidopsis thaliana MERISTEM LAYER1* (*ATML1*; *Roeder et al., 2012*), which encodes a class IV homeodomain-leucine zipper transcription factor (*Lu et al., 1996*; *Nakamura et al., 2006*; *Schrick et al., 2004*). Previous research has indicated that *ATML1* is necessary for establishing the epidermal cell layer during early embryogenesis (*Lu et al., 1996*; *Roeder et al., 2012*; *Sessions et al., 1999*; *Takada and Jürgens, 2007*). Plants doubly mutant for *atml1* and its closely related paralog, *protodermal factor 2*, lack an epidermal layer and are thus seedling lethal (*Abe et al., 2003*; *Ogawa et al., 2015*). Conversely, ectopic expression of *ATML1* results in inappropriate differentiation of *epidermal* cell types in the inner cell layers of cotyledons (*Peterson et al., 2013*; *Takada et al., 2013*). This result suggests that expression of *ATML1* can promote cells to adopt epidermal-specific cell identity in tissues other than the epidermis.

ATML1 is required for the formation of giant cells; however, only a subset of cells expressing *ATML1* become giant in the *Arabidopsis* sepal epidermis. This raises the question of what patterning mechanism could lead to a scattered pattern of giant cells interspersed between smaller cells. Here, we use live imaging, quantitative image analyses and computational modeling to demonstrate that fluctuations in the concentration of the transcription factor ATML1 initiate the pattern of giant and small cells in the *Arabidopsis* sepal.

## Results

### *ATML1* works in a dosage-dependent manner

To determine how ATML1 specifies giant cells when it is expressed in every cell, we overexpressed *ATML1* in the epidermis by approximately five-fold by using the *PROTODERMAL FACTOR1* (*PDF1*) promoter (*pPDF1::FLAG-ATML1*; *Figure 2A and G*; *Abe et al., 2001*, *2003*, *2003*; *San-Bento et al., 2014*). *ATML1* overexpression lines produced sepals almost entirely covered in giant cells

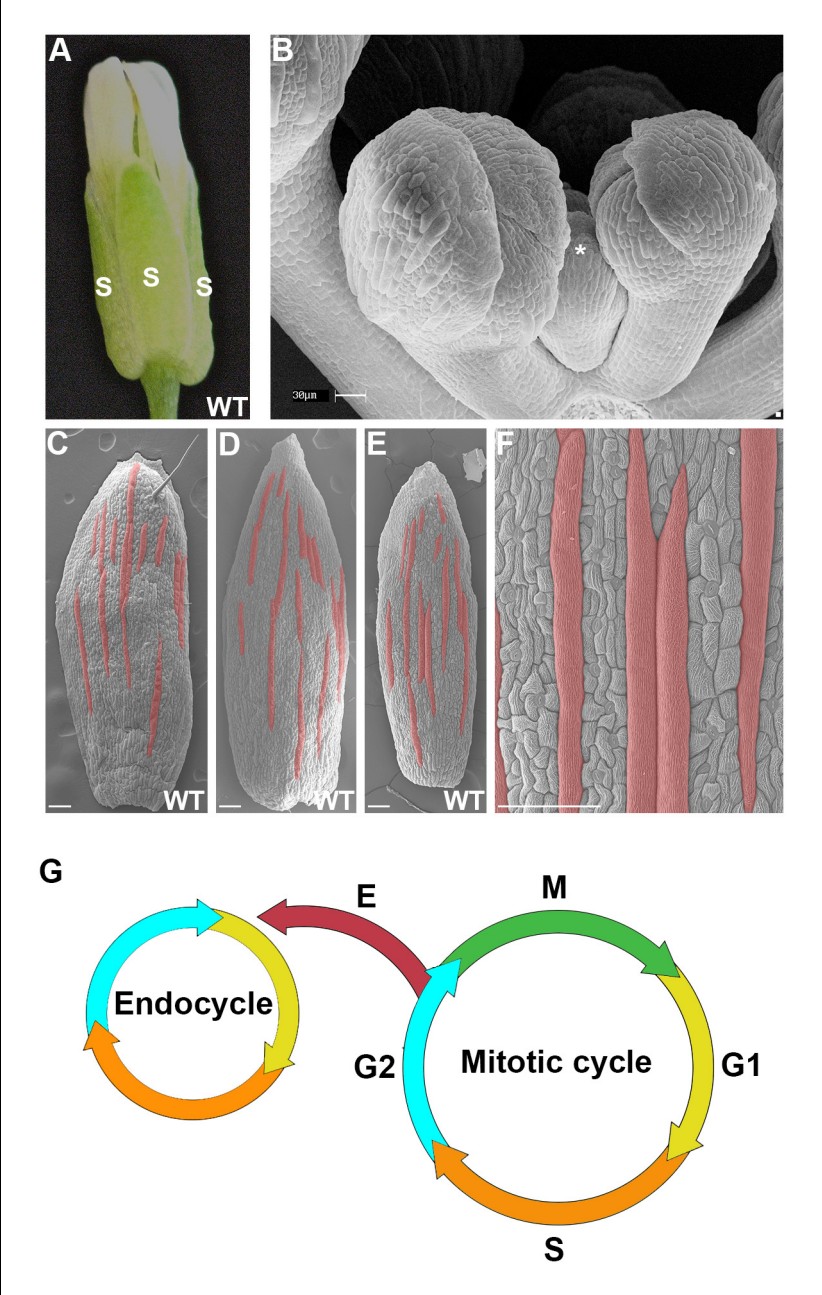

**Figure 1.** The scattered pattern of giant epidermal cells. (**A**) An image of a wild-type (WT) *Arabidopsis thaliana* flower. The sepals (s) are the outermost leaf-like floral organs. (**B**) SEM image of developing sepals on young flower buds. The three flowers in the middle are in approximately the same orientation and stages as the live imaged sepals. Live images typically start with sepals at the youngest stage shown, exemplified by the center flower (*). (**C–F**) SEM images of mature wild-type sepals. Each sepal exhibits variations in the arrangement of giant cells. Giant cells are false colored in red using Photoshop. Magnified view of **E** shown in **F**. Scale bars in **B**, 30 μm and in **C–F**, 100 μm. (**G**) A cell cycle diagram depicting the mitotic cell cycle and the endoreduplication cycle (endocycle). During the mitotic cycle, a new 2C cell will enter Gap 1 (G1). In G1, the cell will increase its size in preparation for DNA synthesis (S), where it will then become 4C. After S phase, the cell will enter Gap2 (G2), where it will continue to grow in size and produce more protein in preparation for mitosis (M). Completion of mitosis will result in the formation of two 2C daughter cells, which will then re-enter the mitotic cycle. Alternatively a cell may endocycle (E), where a cell will go through G1, S, G2 but bypass M to form a polyploid cell. Note that giant cells are 8C and higher polyploid epidermal cells that form through endoreduplication.

(*Figure 2A*). Since giant cells endoreduplicate (16–32C in ploidy; *Roeder et al., 2010*), we tested whether ATML1 overexpression also induced endoreduplication. As expected, the proportion of highly endoreduplicated epidermal nuclei from *ATML1* overexpression line sepals increased (*Figure 2I*, red bars). These sepals contained a greater proportion of 16C and 32C giant cells than wild type, and on occasion a few cells even underwent an additional endocycle (64C; *Figure 2I*, red bars). In addition, we have previously demonstrated that giant and small epidermal cells can be distinguished with two molecular markers (*Roeder et al., 2012*). To test whether our *ATML1* overexpression line sepals confer giant cell identity, we crossed them with plants expressing the giant and small cell markers. In these crossed sepals, the giant cell marker was expressed in almost every epidermal cell and the small cell marker was expressed only in a few remaining small cells (*Figure 2J and K*). To validate that ATML1 alone is sufficient to drive giant cell formation, we induced *ATML1* expression in inflorescences using an ATML1 estradiol-inducible line. Ectopic giant cells formed on the sepal five days after being treated with 10 μM estradiol (*Figure 2—figure supplement 1*). Overall, these results suggest that high levels of ATML1 are sufficient to induce sepal epidermal cells to adopt giant cell identity and can force a deterministic all-giant cell pattern.

Since *ATML1* is expressed in every epidermal cell (*Abe et al., 2003*; *Lu et al., 1996*; *Roeder et al., 2010*, *2012*; *Sessions et al., 1999*) and *ATML1* overexpression leads to an ectopic all-giant cell phenotype, we wondered whether epidermal cell identity specification is sensitive to the dosage of *ATML1*. We altered levels of *ATML1* genetically to test whether that would change the proportion of giant cells in the sepal epidermis (*Figure 2A–F*). First, we reduced levels of ectopic *ATML1* expression by crossing our *ATML1* overexpression line with wild-type plants, resulting in plants containing only one copy of the *ATML1* overexpression transgene. These hemizygous plants formed ectopic giant cells, but fewer than the homozygous overexpression lines, and had more small cells (*Figure 2B*). To reduce ectopic *ATML1* levels further, we crossed *ATML1* overexpression hemizygotes into an *atml1–3* mutant background, removing endogenous *ATML1* expression. This resulted in plants with even fewer ectopic giant cells and more small cells (*Figure 2C*). To test dosage dependency further, we examined *atml1–3* heterozygous mutant plants. These plants had fewer giant cells than WT but more than *atm1–3* homozygous mutants (*Figure 2D,E and F*). We verified through qPCR that inflorescences from each of these *ATML1* dosage genotypes expressed different amounts of *ATML1* as expected (*Figure 2G*). Additionally, we used flow cytometry to quantify endoreduplication and semi-automated image processing to measure cell size (*Figure 2H and I*; *Cunha et al., 2010*; *Roeder et al., 2010*). Each dosage genotype exhibited proportional changes in ploidy and cell size. Together, these results suggest that *ATML1* influences giant cell formation in a dosage-dependent manner, where the amount of *ATML1* expressed will determine the proportion of giant cells that form in the sepal.

## ATML1 levels differ between neighboring sepal cells

The dosage dependency of *ATML1* suggests that the level of *ATML1* expression in each sepal is critical for establishing giant cell and small cell patterning. Furthermore, moderate overexpression of *ATML1* prompts only some cells to become giant (*Figure 2A–F*), suggesting that either cells exhibited varying responses to the same ATML1 concentration or that ATML1 concentrations varied between cells. To quantify ATML1 levels in individual cells during sepal development and distinguish between these possibilities, we created a mCitrine-ATML1 fusion protein reporter (*pATML1::mCitrine-ATML1*) and transformed it into *atml1–3* mutant plants (*Figure 3*). This reporter expresses mCitrine-ATML1 under the putative native *ATML1* promoter and 3' UTR. We recovered two independent transgenic lines that fully rescue the *atml1–3* loss-of-giant cell mutant phenotype (*Figure 3A–D*; Materials and methods). Both lines exhibited similar behavior, thus we focused our analysis on one of them. Overall, these results suggest that our mCitrine-ATML1 fusion protein functions similarly to endogenous ATML1 (*Figure 1C–F*; *Figure 3A–D*).

To quantify mCitrine-ATML1 fluorescence in each epidermal cell of early developing sepals and floral meristems, we developed and implemented an image analysis pipeline (*Box 1*; *Box 1—Figure 1*). We observed that in the developing sepal, mean normalized mCitrine-ATML1 concentrations differ between individual nuclei (*Figure 3G–J*; sepals show a mean coefficient of variation (CV) of approximately 0.2). Conversely in the floral meristem, which does not form giant cells, mCitrine-ATML1 concentrations are more uniform (*Figure 3E–F and I–J*; meristems show a mean CV of approximately 0.1). In particular, we can see that although unimodal, the distribution of ATML1



**Figure 2.** ATML1 levels influence the quantity of giant cells that form on the sepal. (**A–F**) SEM images of sepals from an *ATML1* genetic dosage series. Giant cells are false colored in red. (**A**) *ATML1* overexpression line that is homozygous for the *pPDF1::FLAG-ATML* transgene. (**B**) *ATML1* overexpression line that is hemizygous for the *pPDF1::FLAG-ATML1* transgene. (**C**) *ATML1* overexpression line hemizygous for the *pPDF1::FLAG-ATML1* transgene crossed into a *atml1–3* mutant background. (**D**) Wild type. (**E**) *atml1–3/+* heterozygous mutant. (**F**) *atml1–3* homozygous mutant. (**G**) qPCR on inflorescences from dosage series verifying that *ATML1* mRNA levels vary between lines as expected. Fold change is calculated as the average of three biological replicates. Error bars represent the extended standard deviation. (**H**) Quantification of the average number of giant cells per sepal in *ATML1* dosage series using semi-automated image processing. Giant cells are defined as cells with an area larger than 4000 µm². Error bars represent the standard error of mean, n = 3 sepals per genotype, with each pooled genotype having >1000 cells analyzed. (**I**) Ploidy of epidermal cells in sepals of the *ATML1* dosage series determined by flow cytometry. Inset shows percentage of high ploidy nuclei. Average of 3 biological replicates with >40,000 nuclei analyzed per replicate; error bars represent standard error of mean. Note that epidermal cells include a large number of 2C and 4C cells on the back (adaxial) side of the sepal in all genotypes, which are not affected by *ATML1* overexpression. (**J–K**) Confocal maximum intensity projection image of a wild-type (**J**) and *ATML1* overexpression (**K**) sepal expressing the giant (3xvenus, nuclear localized, blue) and small cell (GFP, ER localized, green) molecular markers. Cell walls are stained with propidium iodide (PI, red). In the *ATML1* overexpression sepal (**K**), the giant cell marker is expressed in almost every cell and the small cell marker is

*Figure 2 continued*

extremely reduced. Note: Margin cells at the edges of the sepals are distinct cell types that are not affected by ATML1. Scale bars in **A**–**F**, 100 μm. T-tests were performed between genetically altered dosage series and wild-type sepals. p-value ≤ 0.05 marked with *, p-value ≤ 0.01 marked with **, and non-significant denoted by ns.

The following figure supplement is available for figure 2:

**Figure supplement 1.** ATML1 estradiol inducible transgenic plants form ectopic giant cells five days after application of 10 μM estradiol.

concentrations in individual nuclei is broader in the sepal than in the meristem, both for lower and higher values (*Figure 3J*). This suggests that ATML1 concentration behaves differently depending on the developmental context. To see whether other genes also exhibit variable expression similarly to mCitrine-ATML1 in the developing sepal nuclei, we measured the expression of two fluorescently-tagged transcription factors, VIP1-mCitrine (*pVIP1::VIP1-mCitrine*) and AP2-2XYpet (*pAP2::AP2-2XYpet*), and the *SEC24A* transcriptional reporter (*pSEC24A::H2B-GFP*). VIP1 is a mechano-sensitive transcription factor that localizes to the nucleus upon hypo-osmotic treatment (*Tian et al., 2004*; *Tsugama et al., 2016*) and AP2 is a master regulator of floral organ identity that is expressed in sepals (*Wollmann et al., 2010*). SEC24A is a ubiquitously expressed CopII vesicle-coat protein that is involved in vesicle trafficking from the ER to the Golgi and has been previously reported to influence giant cell formation on the sepal (*Qu et al., 2014*). We found that mCitrine-ATML1 concentrations in the sepal were approximately twice as variable as the other reporters (*Figure 3I*, *Figure 3—figure supplement 1*; VIP1 sepals show a mean CV of approximately 0.12; AP2 sepals show a mean CV of approximately 0.14; SEC24A sepals show a mean CV of approximately 0.12), suggesting that varying expression levels in sepal epidermal cells is not a common feature observed for every gene.

## Live imaging shows mCitrine-ATML1 fluctuates in developing sepal cells

Since ATML1 levels differ among cells and higher ATML1 levels increase the proportion of giant cells in the sepal, we hypothesized that in wild-type sepals ATML1 levels fluctuate in all epidermal cells, with only some cells passing a threshold to promote giant cell fate. According to this hypothesis, to become a giant cell, a sepal epidermal cell would need to experience a high concentration of ATML1 above a threshold. In contrast, to become a small cell, a sepal epidermal cell would experience only lower concentrations of ATML1 that fall below the threshold while fluctuating.

To determine whether ATML1 fluctuates within single cells, we live imaged the mCitrine-ATML1 reporter in developing sepal primordia every 8 hr until giant cells formed and used our image analysis pipeline to track fluorescence in each nucleus over time (*Figure 4A*; *Figure 4—figure supplements 1A; 2A* and *3A*; *Box 1*; *Box 1—Figure 1*; *Videos 1–4*). We found that during early sepal development, epidermal cells not only have varying amounts of mCitrine-ATML1, but also that mCitrine-ATML1 levels fluctuate within individual cells over time (*Figure 4A–C*; *Figure 4—figure supplements 1A–C; 2A–C;* and *3A–C*).

After specification, giant cells immediately enter endoreduplication during early sepal development and endoreduplicating nuclei can be recognized by their size and shape (*Roeder et al., 2010*). We therefore classified nuclei that start to endoreduplicate and become 8C or higher as giant cell nuclei. We verified this by following giant cell differentiation throughout our live imaging series and by comparing these nuclei to nuclei of giant cells defined by cell size in sepals expressing a plasma membrane marker (*Figure 4—figure supplement 4*).

To assess whether cells destined to be giant have fluctuations of ATML1 that reach higher peak concentrations than cells destined to be small, we tracked mCitrine-ATML1 levels in sepal primordia throughout our live imaging series (*Figure 4C*; *Figure 4—figure supplements 1C*, *2C* and *3C*). We observed that cells that eventually become giant generally exhibit fluctuations reaching higher concentrations of mCitrine-ATML1 before endoreduplication initiates than cells that mitotically divide. However, we observed high fluctuations in some cells that divided to become small cells (*Figure 4C*; *Figure 4—figure supplements 1C*, *2C* and *3C*). To quantitatively determine whether there was an ATML1 concentration threshold that could discriminate between cells that would become giant or cells that would remain small, we assessed how well mCitrine-ATML1 concentration peaks in each

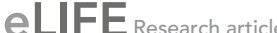

**Figure 3.** mCitrine-ATML1 expression is variable from cell to cell in the sepal but uniform in the meristem. (**A**) SEM image of a wild-type (Col) sepal. (**B**) SEM image of an *atml1–3* mutant sepal. Note that *atml1* mutants exhibit a lack-of-giant-cell phenotype. (**C–D**) SEM images showing that the *pATML1::mCitrine-ATML1* transgene rescues the lack-of-giant-cell phenotype normally exhibited by the *atml1–3* mutant. Additionally, both the number and spacing

*Figure 3 continued on next page*

*Figure 3 continued*

pattern of giant cells appear similar to wild type (**A**). Giant cells in (**A–D**) are false colored red. (**E**) Confocal denoised images of three floral meristems expressing *pATML1::mCitrine-ATML1* (white). (**F**) Heat maps of mean normalized concentration levels of mCitrine-ATML1 expression in the floral meristems. (**G**) Confocal denoised images of three young sepal primordia expressing *pATML1::mCitrine-ATML1* (white) (right most sepal is shown later in *Figure 4—figure supplement 2* as time 0 hrs of the 3rd mCitrine-ATML1 reporter sepal). (**H**) Heat maps of mean normalized concentration levels of mCitrine-ATML1 expression in the young sepal primordia. (**I**) Dot plot of the coefficients of variation (CV) of normalized fluorescent protein concentration in each sample. The CV of mCitrine-ATML1 in nuclei of young developing sepals is higher than in nuclei of floral meristems. The high CV is specific to mCitrine-ATML1 as VIP1-mCitrine (*pVIP1::VIP1-mCitrine*), AP2-2XYpet (*pAP2::AP2-2XYpet*) and a *SEC24A* transcriptional reporter (*SEC24::H2B-mGFP*) have lower CVs in young sepals. n = 3 for each genotype. (**J**) Histograms of normalized mCitrine-ATML1 concentrations for sepals (from **H**; red) and meristems (from **F**; blue). Both histograms show a unimodal distribution, however the distribution of ATML1 concentrations in single cells is broader in the sepal than in the meristem. Scale bars in **A–D** 100 µm; **E** and **G**, 10 µm. The number of cells analyzed for mCitrine-ATML1 meristems from left to right: n = 102, 136 and 82. The number of cells analyzed for each mCitrine-ATML1 sepal primodium in order from left to right: n = 91, 48 and 142. Denoised images and corresponding heat maps for *pSEC24A::H2B-GFP*, VIP1-mCitrine and AP2-2XYpet sepals are shown in *Figure 3— figure supplement 1*.

The following figure supplement is available for figure 3:

**Figure supplement 1.** The transcriptional reporter *SEC24A:: H2B-GFP* and the fusion proteins VIP1-mCitrine, and AP2-2XYpet are uniformly expressed in the developing sepal.

cell lineage were able to discriminate between giant cell and small cell fate. To do this, we measured the peak concentration of mCitrine-ATML1 in cells that either go on to divide (small) or endoreduplicate (giant) and performed a receiver operator characteristics (ROC) analysis using these two classes (*Figure 4D and E*; *Figure 4—figure supplements 1D and E*, *2D and E* and *3D and E*; *Chao et al., 2015*; *Schröter et al., 2015*; *Teles et al., 2013*). In this type of analysis, the ratio of correctly and incorrectly classified cells (i.e. the true positive rate (TPR) and false positive rate (FPR)) is calculated for a varying threshold value, providing a characteristic curve. The area under this curve (AUC) provides a measure of accuracy for predicting cell fate based on ATML1 concentration peaks (1 being perfect and 0.5 no better than random classification). We observed an average AUC of 0.74 in our different datasets, highlighting the predictive power of ATML1 concentration peaks in discriminating small versus giant cell fate (AUC = 0.76, 0.69, 0.73, 0.78; *Figure 4E*; *Figure 4—figure supplements 1E*, *2E* and *3E*). Additionally, for each case we were then able to infer an optimum ATML1 concentration threshold that provides maximum separation between the cells that become giant and cells that remain small, i.e. the concentration value that maximizes the difference between TPR and FPR. We considered this threshold to be indicative of the ATML1 concentration required to trigger endoreduplication for the majority of cells in a given sepal.

In summary, we show that the heterogeneity in ATML1 among cells in the sepal primordium can be explained by dynamic cell-autonomous fluctuations, where giant and small cell fate are strongly correlated with the concentration of ATML1 reached. Cells with high concentration fluctuations of ATML1 will likely endoreduplicate and become giant, whereas cells with low concentration fluctuations will likely go on to divide and remain small.

## G2 phase of the cell cycle gates specification of giant cells

Since the decision to endoreduplicate causes a cell to bypass mitosis (*Figure 1G*; *Inzé and De Veylder, 2006*; *Sugimoto-Shirasu and Roberts, 2003*), we wondered whether high levels of ATML1 needed to occur at a particular stage of the cell cycle to modulate cell-fate decisions. It has been previously demonstrated that in *Arabidopsis* there is a linear correlation between nuclear size and cell ploidy (*Jovtchev et al., 2006*). Using our live imaging data, we therefore characterized cell cycle stages by ploidy at each time point, using nuclear size as a proxy, where 2C is associated with cells being in G1 and 4C is associated with cells being in G2 (See *Box 2* and Material and methods for ploidy determination). Next, we compared peak concentration levels of mCitrine-ATML1 in individual cell lineages during both the 2C and 4C ploidy states of the cell cycle immediately before entry

## Box 1. mCitrine-ATML1 image quantification and tracking pipeline.

We designed and implemented an image analysis pipeline to quantify the concentration of mCitrine-ATML1 in individual epidermal nuclei, as well as nuclear size and shape parameters, while simultaneously tracking each cell lineage during sepal development. Raw intensity images were filtered for Poisson-Gaussian mixed noise using the ImageJ plugin PureDenoise (**Box 1—Figure 1A and B**; **Blu and Luisier, 2007**; **Luisier et al., 2009, 2010**). The resulting denoised images were imported into MorphoGraphX (**Barbier de Reuille et al., 2015**) and used as input for binary mask creation (**Box 1—Figure 1C**). The purpose of the binary mask is to separate sepal epidermal cells from background noise and underlying cell layers during the quantification step. Finally, the binary masks were imported into Costanza (http://www.plant-image-analysis.org/software/costanza) in order to perform segmentation of each individual nucleus (**Box 1—Figure 1D**).

To spatiotemporally track individual nuclei, Canny edge detection (https://imagej.nih.gov/ij/plugins/canny/index.html) was initially performed by applying the FeatureJ ImageJ plugin (http://www.imagescience.org/meijering/software/featurej/) to each denoised image, facilitating the subsequent registration step. Pairing of individual nuclei in two consecutive time points was computed by registering pairs of successive images (**Box 1—Figure 1E**; **Commowick et al., 2008**; **Michelin et al., 2016**; **Ourselin et al., 2000**) and then computing the optimal cell-cell pairing using ALT (**Fernandez et al., 2010**). In order to ensure that all nuclei were correctly tracked, successive image pairs were imported into MorphoGraphX along with the associated nuclei pairings provided by ALT and incorrectly tracked or unlabeled nuclei were manually corrected using the parent labels tool (**Box 1—Figure 1F**).

Raw intensity and nuclear segmentation images, as well as the corrected parental correspondence tables, were imported into an in-house developed MATLAB quantification module, for statistical analysis. For each nucleus, this module selected the slice with the largest area and quantified total fluorescence intensity within this slice from the raw intensity image (**Box 1— Figure 1G**). For each cell, in every time point, concentrations (**Box 1—Figure 1H**), areas and nuclear shape parameters were quantified. Nuclear pairing tables between consecutive time points were used to establish cell lineages for each time course, and each of the variables could then be tracked in time for each time course of sepal growth (e.g. **Box 1—Figure 1I**).

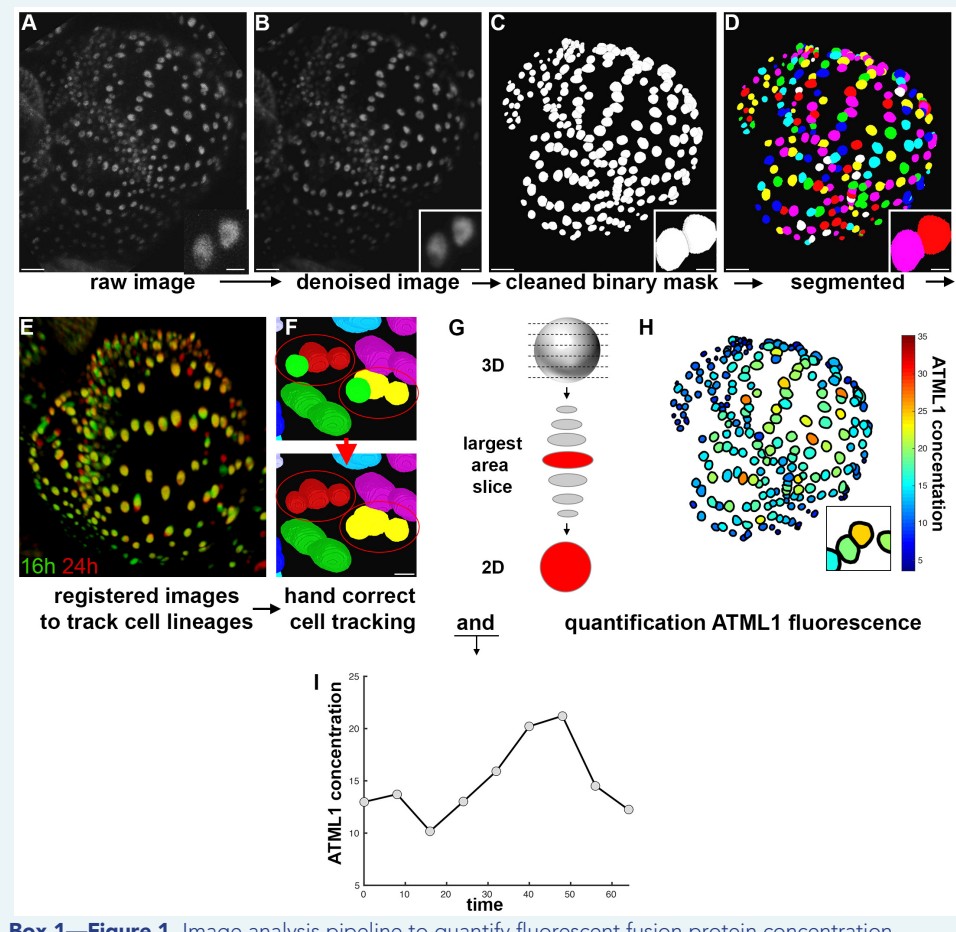

**Box 1—Figure 1.** Image analysis pipeline to quantify fluorescent fusion protein concentration.

Box 1—Figure 1 continued

(A) Raw confocal image of developing sepal expressing mCitrine-ATML1 (sepal also presented in *Figure 4*). (B) Denoised confocal images using PureDenoise ImageJ software. (C) Binary mask created in MorphoGraphX. (D) Segmented image created in Costanza. (E) 3D projection of registered pairs of consecutive sepal confocal acquisitions (16 hr in green and 24 hr in red). (F) Manual correction of incorrectly tracked nuclei in MorphoGraphX. Top panel shows two examples where ALT did not correctly track one of two daughter cells. Bottom panel shows that nuclei can be manually corrected in MorphoGraphX. (G) Schematic of quantification process. A MATLAB module detects the confocal z-stack slice with largest area for each nucleus. Then, fluorescence concentration is quantified (total fluorescence divided by area) using the raw intensity z-stack. (H) Heat map of the fluorescence concentration for each nucleus on the sepal. (I) Example of ATML1 fluorescence concentration in one nucleus tracked through time.

into either mitosis or endoreduplication (*Figure 4F–I*; *Figure 4—figure supplements 1F–I*, *2F–I* and *3F–I*). We found that in the preceding cell cycle, both small cells and giant cells show similar peak levels of mCitrine-ATML1 in 2C (*Figure 4F*; *Figure 4—figure supplements 1F*, *2F* and *3F*). Our ROC analysis shows that ATML1 concentration peaks during the G1 (2C) stage are not predictive of cell fate (AUCs = 0.54, 0.37, 0.43, 0.37; *Figure 4G*; *Figure 4—figure supplements 1G*, *2G* and *3G*). In contrast, most cells that experience relatively high peak concentrations of mCitrine-ATML1 while in 4C endoreduplicate and become giant cells (*Figure 4H,J–Q*; *Figure 4—figure supplements 1H,J–Q*, *2H,J–O* and *3H,J–Q*). Our ROC analysis is consistent with this observation, showing that ATML1 concentration peaks in 4C are strongly predictive of cell fate (AUCs = 0.80, 0.80, 0.80, 0.84; *Figure 4I*; *Figure 4—figure supplements 1I*, *2I* and *3I*).

Overall, these results suggest that a cell is competent to respond to high levels of ATML1 mainly during G2 to induce giant cell formation.

## Threshold-based mechanism is consistent with increased giant cell formation in *ATML1* overexpression lines

Given that high ATML1 levels during the G2 stage of the cell cycle are associated with giant cell formation, we wondered whether all epidermal cells were expressing ATML1 above the giant cell threshold in our *ATML1* overexpression sepals to produce an ectopic giant cell phenotype. To address this question, we live imaged early sepal development every 8 hr in plants that had GFP-ATML1 expressed under the PDF1 promoter, which produce the ectopic giant cell phenotype (*Figure 5A*; *Figure 5—figure supplements 1A* and *2A*; *Videos 5*, *6* and *7*). As expected, for a promoter with an ATML1 binding site, PDF1::GFP-ATML1 levels fluctuated in individual cells (*Figure 5B and C*; *Figure 5—figure supplements 1B, G*, *2B and G*).

We next tested whether most epidermal cells surpassed the ATML1 threshold in G2 to induce endoreduplication. Since very few cells divide in our *pPDF1::GFP-ATML1* sepals, we could not directly infer this threshold through ROC analysis from this data as before. Therefore, we derived a common ATML1 concentration threshold from the live imaging data of our *pATML1::mCitrine-ATML1; atml1–3* flowers (*Figure 4—figure supplement 5*), by performing ROC analysis using mean normalized ATML1 concentrations for each flower (see Materials and methods for details). Applying this threshold to the *pPDF1::GFP-ATML1* data, we observed that almost all endoreduplicating cells exhibited high peak levels of GFP-ATML1 in G2, above the common threshold (*Figure 5D–J*; *Figure 5—figure supplements 1C–F*; and *2C–F*). This is in contrast to wild type, where fewer cells reach the ATML1 concentration threshold (*Figure 4*). Combined, these data suggest that our overexpression line follows the same threshold-based cell-autonomous fluctuation patterning mechanism; the increased basal *GFP-ATML1* expression from the *PDF1* promoter raises ATML1 production levels such that almost all sepal epidermal cells surpass the giant cell fate-inducing threshold during G2.

## The dynamics of ATML1 fluctuations are independent of LGO and endoreduplication

We have previously published that a cyclin dependent kinase inhibitor, LOSS OF GIANT CELLS FROM ORGANS (LGO), is required for giant cell formation; LGO triggers endoreduplication once giant cell fate has been established (*Roeder et al., 2012*). To verify that LGO acts genetically

**Figure 4.** ATML1 fluctuates in sepal epidermal cells to initiate giant cell patterning. (A) Raw images of *pATML1::mCitrine-ATML1* (white) from a live imaging series of a developing sepal. Images were taken every 8 hr for 64 hr. (B) Heat map showing corresponding mCitrine-ATML1 concentrations (total fluorescence divided by area) at each time point from (A). (C) mCitrine-ATML1 concentrations tracked over time in cells that became giant (red) and cells that divided to stay small (blue). (D) mCitrine-ATML1 peak concentration levels in each lineage preceding endoreduplication or mitotic

*Figure 4 continued on next page*

*Figure 4 continued*

division (Materials and methods). The concentration threshold that best separates giant cells from small cells is shown as a dashed line. (**E**) Receiver operating characteristic (ROC) curve (red) for (**D**). The ratio of correctly and incorrectly classified cells (i.e. the true positive rate (TPR) and false positive rate (FPR)) is calculated for a varying threshold value, providing a characteristic curve. The area under the curve (AUC) provides a measure of accuracy for predicting cell fate based on ATML1 concentration (1 being perfect and 0.5 no better than random classification). The AUC is 0.76. The black dot marks the optimal concentration threshold where the difference between TPR and FPR is maximal. (**F–I**) mCitrine-ATML1 peak concentrations and ROC analysis for G1 (2C) or G2 (4C) phases of the cell cycle preceding endoreduplication or mitotic division. (**F**) mCitrine-ATML1 peak concentration levels and optimal concentration thresholds separating giant cells from small cells at G1. (**G**) ROC curve for (**F**). (**H**) mCitrine-ATML1 peak concentration levels and optimal concentration thresholds separating giant cells from small cells at G2. (**I**) ROC curve for (**H**). For (**G**) AUC = 0.52 (not predictive) and for (**I**) AUC = 0.8 (predictive of cell fate). (**J–M**) Single cell lineages tracked through time (64 hr). Each denoised nucleus image is outlined in a color associated with its ploidy: yellow = 2C, blue = 4C, and red = 8C and higher. (**J–K**) giant cell and (**L–M**) small cell lineages. (**N–Q**) Tracked mCitrine-ATML1 concentration levels corresponding to the single cell lineages in (**J–M**). The ploidy at each point corresponds to the color of the dot, as above. mCitrine-ATML1 concentrations for all other cell lineages are plotted in grey for context. Note that giant cells in **N** and **O** cross the threshold while they are in G2 (4C) of the cell cycle, while in **Q**, mCitrine-ATML1 crosses the threshold in 2C at t = 48 hr but then the cell goes onto divide. Additionally, the fate of the cell that crosses the threshold in 4C at t = 48 hr remains unknown. A total of 110 lineages were analyzed (n = 646 cells). This flower is shown in *Video 1*. Three similar replicate flowers are shown in the *Figure 4—figure supplements 1*, *2* and *3*.

The following figure supplements are available for figure 4:

**Figure supplement 1.** Second flower that demonstrates ATML1 fluctuates in sepal epidermal cells to initiate giant cell patterning.

**Figure supplement 2.** Third flower that demonstrates ATML1 fluctuates in sepal epidermal cells to initiate giant cell patterning.

**Figure supplement 3.** Fourth flower that demonstrates ATML1 fluctuates in sepal epidermal cells to initiate giant cell patterning.

**Figure supplement 4.** Giant cells can be identified by their large, elongated, endoreduplicating nuclei.

**Figure supplement 5.** Mean normalized mCitrine-ATML1 concentrations for all four *pATML1::mCitrine-ATML1;atml1–3* flowers.

---

downstream of ATML1 to establish giant cells, we crossed our ATML1 overexpression line (*pPDF1:: FLAG-ATML1*) to our *lgo-2* mutant, which exhibits a loss-of-giant cell phenotype (*Figure 6C*). Plants homozygous for both the *lgo-2* mutation and the overexpression transgene do not form giant cells, demonstrating that LGO activity is required downstream of ATML1 for formation of giant cells.

Since LGO acts downstream of ATML1, we hypothesized that ATML1 fluctuations should be unaltered in the *lgo-2* mutant, which fail to endoreduplicate in early stage sepals. In this scenario, we would expect the same number of *lgo-2* nuclei to surpass the ATML1 threshold in G2 as in wild type. Cells that pass the threshold would still divide because they are unable to endoreduplicate. To

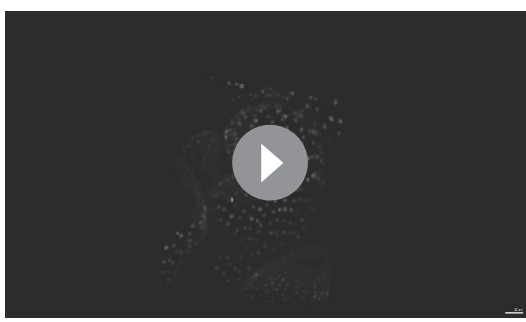

**Video 1.** A movie of a developing *pATML1::mCitrine-ATML1; atml1-3* sepal shown in *Figure 4*. The sepal primordium was live imaged every 8 hr until giant cells form.

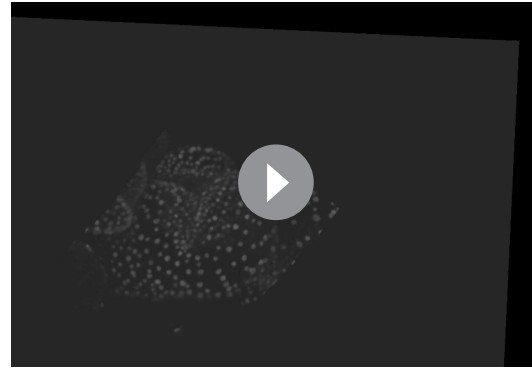

**Video 2.** A movie of a developing *pATML1::mCitrine-ATML1; atml1-3* sepal shown in *Figure 4—figure supplement 1*. The sepal primordium was live imaged every 8 hr until giant cells form.

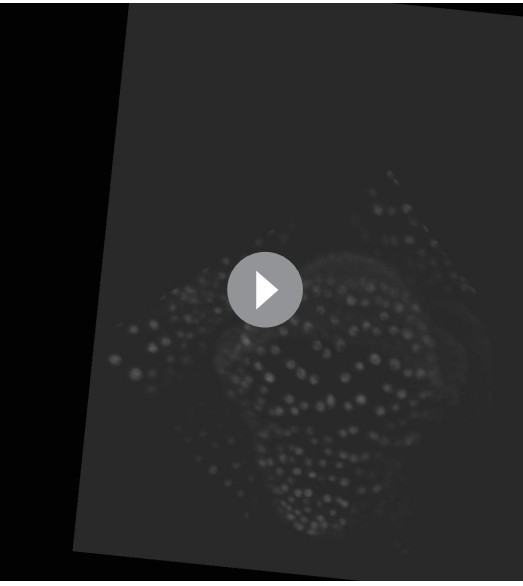

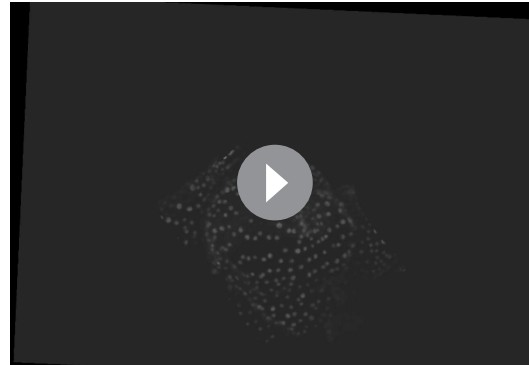

**Video 4.** A movie of a developing *pATML1::mCitrine-ATML1; atm1l-3* sepal shown in *Figure 4—figure supplement 3*. The sepal primordium was live imaged every 8 hr until giant cells form.

**Video 3.** A movie of a developing *pATML1::mCitrine-ATML1; atm1l-3* sepal shown in *Figure 4—figure supplement 2*. The sepal primordium was live imaged every 8 hr until giant cells form.

test this, we live imaged our mCitrine-ATML1 reporter in the *lgo-2* mutant background (*Figure 6A*; *Figure 6—figure supplements 1A* and *2A*; *Videos 8*, *9* and *10*). These plants still exhibited mCitrine-ATML1 fluctuations, suggesting that ATML1 fluctuates independently of LGO (*Figure 6B,E–J*; *Figure 6—figure supplements 1B–F;* and *2B–F*). We applied the common ATML1 concentration threshold derived from *pATML1::mCitrine-ATML1; atml1–3* flowers (see previous section; *Figure 4—figure supplement 5*) to predict the number of giant cells that would have formed exclusively based on the threshold mechanism (ATML1 concentration peaks above threshold during G2; Materials and methods). We found no significant differences between the predicted number of giant cells in the *lgo-2* mutant and the observed number of giant cells in wild type (*Figure 6D–F*; *Figure 6—figure supplements 1C–D* and *2C–D*). This suggests that a cell may still fluctuate to high levels of ATML1 in G2 but without LGO, cells cannot respond to these fluctuations to trigger endoreduplication. Since the absence of LGO does not seem to change the dynamics of ATML1, this result further indicates that ATML1 fluctuations are independent of endoreduplication.

## A model with stochastic fluctuations of ATML1 reproduces giant cell patterning

Previous studies have suggested that gene expression is inherently stochastic, where genes will experience random fluctuations in the rate in which they are transcribed and/or translated (*Elowitz et al., 2002*; *Kaern et al., 2005*). We therefore asked whether a simple computational model that exhibits cell-autonomous stochastic fluctuations of ATML1 is sufficient to recapitulate giant cell patterning as observed in our experimental data. In our model, we implemented a simplified regulatory network, where ATML1 stochastically fluctuates in a growing tissue (*Figure 7A and B*; Materials and methods). In this model, we assume that in every cell there is a basal amount of ATML1 being produced as well as an amount being linearly degraded. In addition, we tested the possibility that ATML1 engages in a self-catalytic feedback loop, as ATML1 has a putative ATML1 binding site in its own promoter and ATML1 has been shown to bind this motif *in vitro* (*Abe et al., 2001*; *Takada and Jürgens, 2007*). Additionally, in seedlings induction of ectopic ATML1 activity for seven days shows an increase of endogenous ATML1 expression 1.5 to two fold, hinting at the possibility of a feedback loop (*Takada et al., 2013*).

# Box 2. Determination of ploidy/cell cycle stage using cell size and shape parameters.

Given the limitations in applying current standard techniques simultaneously with live imaging procedures, we developed a new method to determine ploidy of individual cells throughout live imaging time courses. We used nuclear area as a proxy for defining cell cycle stage since nuclear area and ploidy have previously been described to be linearly correlated in *Arabidopsis* (*Jovtchev et al., 2006*). To confirm this correlation in our sepals, we stained nuclei with DAPI (a chromatin stain previously used to determine ploidy, [*Jovtchev et al., 2006*; *Roeder et al., 2010*]) and subsequently measured nuclear area using ImageJ. We found that there is a linear correlation between nuclear area and ploidy, where nuclear area increases as the cell progresses through the cell cycle, and we could locate discrete area cutoffs that accurately separate different cell cycle stages (2C = G1, 4C = G2; *Box 2—Figure 1A*). Therefore, we propose that this method can in principle be applied to any system in which ploidy has been verified to be linearly correlated with nuclear area.

To further validate that nuclear area correlates with cell cycle progression, we live imaged developing sepals every hour until cells divided (*Box 2—Figure 1F–I*; *Box 2—Videos 1* and *2*). We found that in our *pATML1::mCitrine-ATML1; atml1–3* transgenic plants, individual cells increase their nuclear area to approximately 35 $\mu m^2$ before division. Each resulting daughter cell's nuclear area immediately drops to approximately 15 $\mu m^2$ and then begins to increase its area as the cell progresses through the cell cycle (*Box 2—Figure 1J–Q*). In our observations, mCitrine-ATML1 concentrations do not always exhibit the same trends as area, suggesting that nuclear area is not strongly dependent on mCitrine-ATML1 concentration.

Building on these results which show that area thresholds can be used to effectively separate cell cycle stages, we defined a set of area and eccentricity thresholds to classify cells into different ploidies (2C, 4C, 8C; *Box 2—Figure 1B–E*). For our *pATML1::mCitrine-ATML1 atml1–3* transgenic line, nuclei with an area of <35 $\mu m^2$ were classified as 2C (G1), nuclei with an area of ≥35 $\mu m^2$ with an eccentricity of ≤0.7 were classified as 4C (G2) and nuclei with an area of >35 $\mu m^2$ with an eccentricity of >0.7 were classified as 8C (endoreduplicating). Nuclei that bordered these area thresholds were manually checked to ensure that they were correctly classified, with a small number of incorrectly classified nuclei being reclassified. Manual correction was based on additional knowledge from the live imaging time series and visualization in 3D (e.g. the existence of incorrect transitions such as 2C to 8C or 8C to 2C, known not to happen in normal sepal development). Importantly, information of ATML1 concentration values was not used for ploidy classification at any stage. Additionally, independent manual correction of ploidy classification by different researchers produced highly similar results. For other genotypes (i.e. *pATML1::mCitrine-ATML atml1–3 lgo-2*, *pPDF1::GFP-ATML1*), area and eccentricity threshold values were slightly adjusted in order to account for changes in segmentation parameters (Materials and methods). Flowers that have a broader distribution of giant and small cells tended to have slightly inflated segmented masks in order to increase the number of nuclei successfully segmented through the entire time course. The inflation of the segmented masks leads to slightly increased nuclear area, which we accounted for when we defined the thresholds.

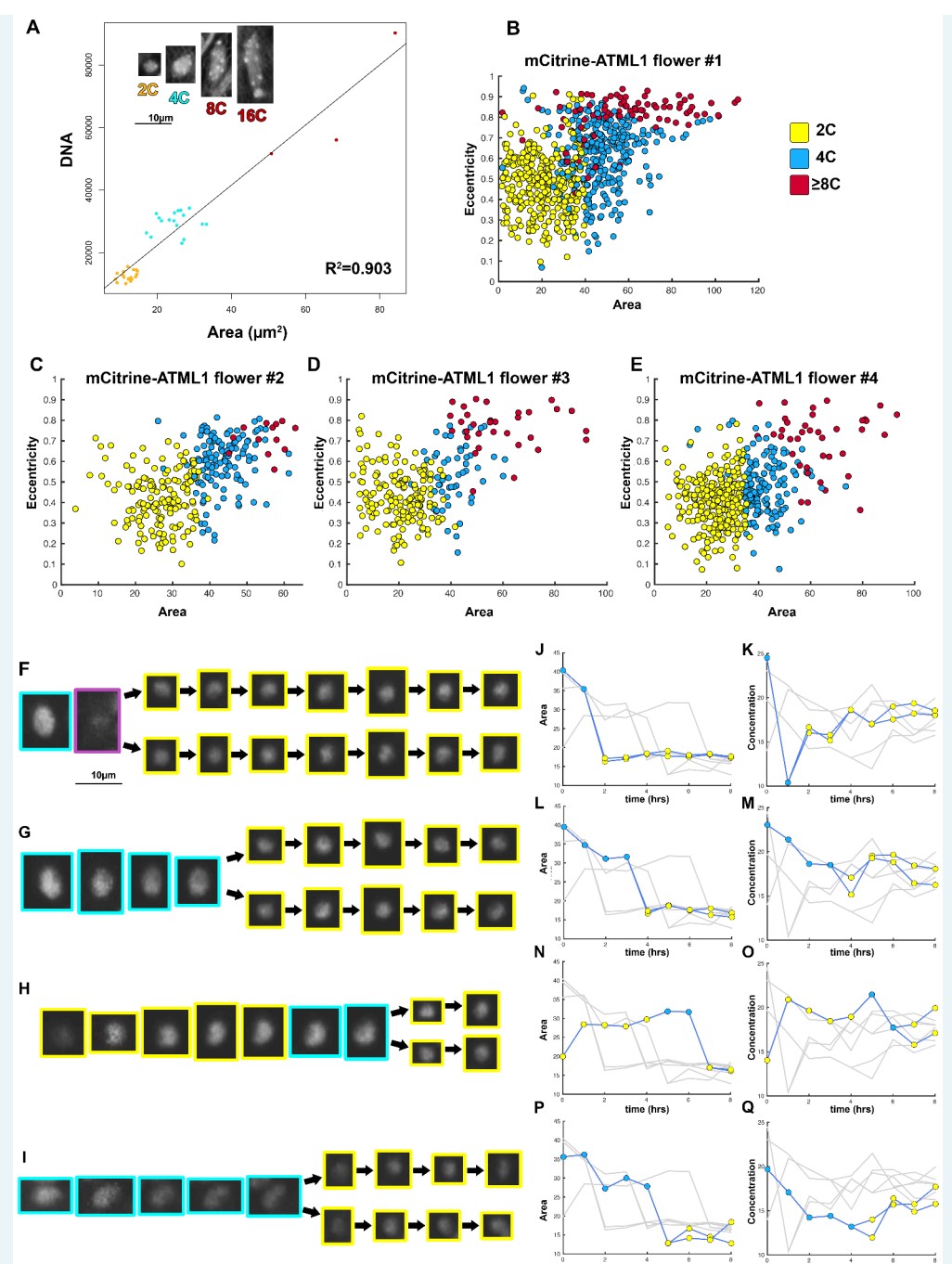

**Box 2—Figure 1.** Nuclear area was used to determine cell cycle stage.

(A) DAPI stained wild-type sepal nuclei show that DNA content and nuclear area are linearly correlated ($R^2$ = 0.903). 2C nuclei are colored yellow, 4C nuclei are colored blue, and 8C/16C nuclei are colored red. One representative confocal image of each classified nucleus is inset on the top left of the graph. Scalebar = 10 $\mu m^2$. N = 38 nuclei were analyzed. (B–E) Area versus eccentricity of different ploidies classified from an area threshold using *pATML1::mcitrine-ATML1;atml1–3* flowers. 2C cells in yellow are <35 $\mu m^2$ in area. 4C cells are in blue and are ≥35 $\mu m^2$ in area with an eccentricity of ≤0.7. Endoreduplicating cells (≥8C) are >35 $\mu m^2$ with an eccentricity of >0.7. In a few instances, a giant cell was poorly segmented and received a low area. These cells were manually corrected. (B) Flower 1; a total of n = 646 cells were analyzed (C) Flower 2; a total of n = 413 cells were analyzed. (D) Flower 3; a total of n = 195 cells were analyzed. (E) Flower 4; a total of n = 436 cells were analyzed. (F–I) Nuclei that undergo a mitotic division from a one-hour interval live imaging series, showing the size change from 4C to 2C after division. (J, L, N, P) Traces of nuclear areas over time corresponding to (F–I). Note that nuclei have an area of approximately 35 $\mu m^2$ before dividing. Immediately upon division, nuclei have an area of approximately 15 $\mu m^2$. (K, M, O, Q) mCitrine-ATML1 concentration of nuclei in (F–I). Note that mCitrine-ATML1

*Box 2—Figure 1 continued*

concentration seemingly fluctuates, independently of nuclear area.

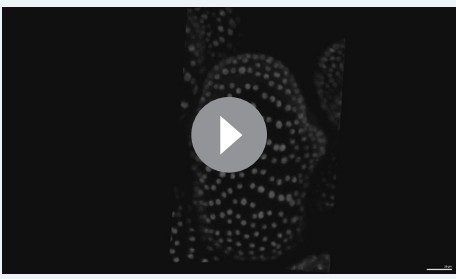

**Box 2—Video 1.** A movie of a developing *pATML1::mCitrine-ATML1; atml1–3* sepal.
The sepal primordium was live imaged every hour to capture the size (area) of nuclei before and after division.
Associated with *Box 2*.

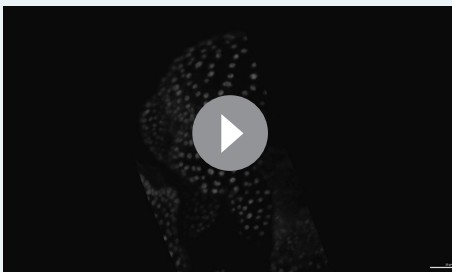

**Box 2—Video 2.** A movie of a developing *pATML1::mCitrine-ATML1; atml1–3* sepal.
The sepal primordium was live imaged every hour to capture the size (area) of nuclei before and after division.
Associated with *Box 2*.

ATML1 is a transcription factor that regulates the expression of downstream genes. Therefore, to induce endoreduplication, ATML1 likely directly or indirectly regulates the expression of a downstream cell cycle regulator (e.g. cyclin/CDK/cyclin-dependent kinase inhibitor). We therefore assigned ATML1 to activate a downstream target that inhibits cell division and promotes entry into endoreduplication. Only if the downstream target passes its own specific threshold in G2, does it successfully drive a cell to endoreduplicate to form a giant cell (*Figure 7A,C*; *Figure 7—figure supplement 1*). Hence, we expect a few cells to divide even if their ATML1 concentrations go above the threshold because the target's threshold is not reached. This is consistent with our live imaging data, where in some cases mCitrine-ATML1 concentrations exceed the giant cell threshold in 4C but the cells go on to divide (*Figure 4—figure supplement 2N*). Furthermore, we expect that a few giant cells will form when ATML1 approaches but does not exceed the threshold because the target stochastically passes its own threshold (*Figure 7—figure supplement 1C*). These circumstances create what we term a soft ATML1 threshold (*Figure 7A*).

In the model, different ploidy and cell division checkpoints were determined using a linearly increasing timer variable, which represents the cell cycle. The timer resets at every cell division checkpoint with a small amount of noise (*Figure 7A*; Material and methods; *Figure 7—figure supplement 1A,B*; *Video 11*).

The model qualitatively reproduced our experimental data and led to a scattered pattern of giant cells in a growing tissue (*Figure 7B*, *Video 11*). Specifically, dynamic fluctuations in ATML1 and in



**Figure 5.** A threshold-based mechanism is consistent with increased giant cell formation in *ATML1* overexpression lines. (A) Raw images of *pPDF1::GFP-ATML1* (white) from a live imaging series of a developing overexpression sepal. Images were taken every 8 hr for 48 hr. (B) Heat map showing corresponding GFP-ATML1 concentrations (total fluorescence divided by area) at each time point from (A). (C) normalized GFP-ATML1 concentrations tracked over time. Note that all cells tracked become giant. (D) Normalized GFP-ATML1 peak concentration levels in each lineage preceding
*Figure 5 continued on next page*

*Figure 5 continued*

endoreduplication for all three *pPDF1::GFP-ATML1* flowers. Dashed line represents the common normalized threshold derived from *pATML1::mCitrine-ATML1;atml1–3* flowers (**Figure 4—figure supplement 5**). Note that almost all nuclei reach high concentrations of GFP-ATML1 above the threshold before endoreduplicating. (E–G) Single giant cells tracked through time (48 hr). Each denoised nucleus image is outlined in a color associated with its ploidy: yellow = 2C, blue = 4C, and red = 8C and higher. (H–J) Tracked normalized GFP-ATML1 concentration levels corresponding to the single cell lineages in (E–F). The ploidy at each point corresponds to the color of the dot, as above. GFP-ATML1 concentrations for all other cell lineages are plotted in grey for context. Note that the giant cells cross the threshold while they are in G2 (4C) of the cell cycle. A total of 23 lineages were analyzed (n = 129 cells). This flower is shown in *Video 5*. Two similar replicate flowers are shown in the *Figure 5—figure supplements 1* and *2*.

The following figure supplements are available for figure 5:

**Figure supplement 1.** Second flower demonstrating that a threshold-based mechanism is consistent with increased giant cell formation in *ATML1* overexpression lines.

**Figure supplement 2.** Third flower demonstrating that a threshold-based mechanism is consistent with increased giant cell formation in *ATML1* overexpression lines.

the target during G2 enable a subset of cells from the developing tissue to become giant cells (*Figure 7C–E*; *Figure 7—figure supplement 1*). We found parameter values that produced wild-type-like sepals, in which the distributions of ATML1 levels and the number of giant cells were similar to those observed experimentally (*Figures 3I–J* and *7D*; Materials and methods). Furthermore, lowering the intensity of the stochastic fluctuations in the model prevented it from matching the experimental data (*Figure 7—figure supplement 2*).

To test whether our model could recapitulate G2-mediated giant cell fate specification, we performed a ROC analysis on the simulated time traces, mimicking the analysis performed on the experimental data (*Figure 7F–G* and *Figure 7—figure supplement 3*). Consistent with our experimental observations, we found lower AUC values in 2C stages than in 4C. This supports our hypothesis that ATML1 levels during the G2 phase of the cell cycle are important for giant cell fate commitment (*Figure 7F–G* and *Figure 7—figure supplement 3A–E*). To further study whether our model could recapitulate our experimental data, in which some fluctuations might be missed due to the 8 hr interval live imaging, we tested whether our AUC analysis would still give similar results when studying the simulated time traces with lower time resolution. We therefore subsampled our simulated data to generate coarse time series, with 80 times lower time resolution than the simulated time step, and we still detected the same trends (*Figure 7—figure supplement 3F–J*).

As previously mentioned, ATML1 might act in a positive feedback loop. We therefore explored different feedback strengths in the parameter space to determine the robustness of our model. We modeled the different feedback strengths by varying the ratio between *ATML1* dependent and basal production rates, whilst keeping the number of predicted giant cells close to experimental values (Materials and methods). With no feedback or low feedback strengths, we could qualitatively match the experimental ROC analysis (*Figure 7—figure supplements 3K–N;* and *4A–B*). In contrast, we were unable to match our experimental data with high feedback strengths because AUC values were predictive of giant cell identity in both 2C and 4C (*Figure 7—figure supplements 3K–N; and 4C–D*). Higher feedback strengths lead to bistability in the system, inducing large and slow fluctuations between high and low levels (*Figure 7—figure supplement 4C–D*).

To test the type of feedback of ATML1 on itself, we examined the effects of induction of *ATML1* on the transcription of the endogenous *ATML1* gene in inflorescences using qPCR (*Peterson et al., 2013*; *Takada et al., 2013*). We

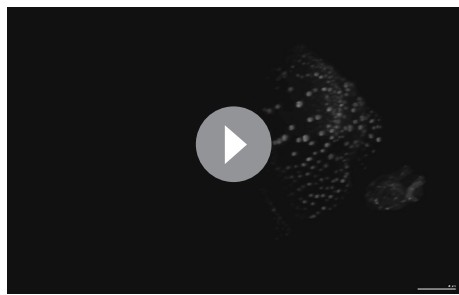

**Video 5.** A movie of a developing *pPDF1::GFP-ATML1* sepal shown in *Figure 5*. The sepal primordium was live imaged every 8 hr until giant cells form.

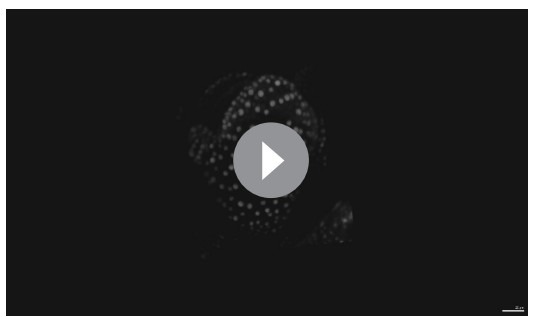

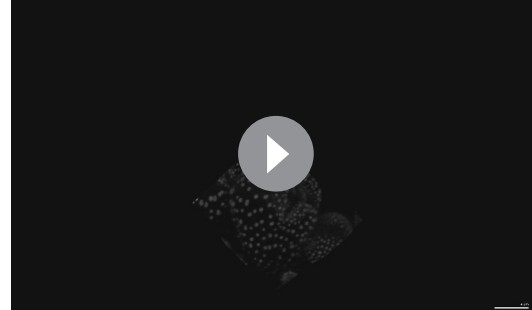

**Video 6.** A movie of a developing *pPDF1::GFP-ATML1* sepal shown in *Figure 5—figure supplement 1*. The sepal primordium was live imaged every 8 hr until giant cells form.

**Video 7.** A movie of a developing *pPDF1::GFP-ATML1* sepal shown in *Figure 5—figure supplement 2*. The sepal primordium was live imaged every 8 hr until giant cells form.

found that *ATML1* induction with 10 µM estradiol lead to total *ATML1* levels 7.1 times higher than the mock treated samples, and increased endogenous *ATML1* expression 1.5-fold within 48 hr (*Figure 7—figure supplement 4E–F*). This level of induction was similar to that observed in other downstream genes, suggesting that the feedback of ATML1 on itself is not activating *ATML1* further than other targets at the 48 hr time point (*Figure 7—figure supplement 4E–F*). The results are also consistent with a previous study carried out in seedlings after 7 days, where endogenous *ATML1* levels increased to 1.7-fold after induction (*Takada et al., 2013*).

To further test the properties of the feedback, we also induced with 0.1 µM or 1 µM estradiol and achieved intermediate levels of induction and activation of downstream genes. In our strong feedback simulations, the parameters chosen are on, or close to, the bistability region in the system, leading to a long-tailed or bimodal distribution of ATML1 expression (*Figure 7—figure supplement 4C–D*), which we do not observe experimentally (*Figure 3J*). Our experimentally observed gradual increase in induced *ATML1* with increasing levels of estradiol further supports the case for weak feedback in the system, as endogenous ATML1 levels are not sensitive to small increases in exogenous *ATML1*. In the strong feedback case, sensitivity to *ATML1* induction increases as the system is bistable and easily reaches the high value state. Thus, our results are consistent with weak feedback in the system.

In order to confirm that endoreduplication can occur only if the target reaches a threshold in G2, we simulated a simpler model where cells could commit to endoreduplication if the target reaches its threshold at any point throughout the cell cycle. In contrast to our experimental data, these simulations led to ATML1 exhibiting high AUC values in both 2C (G1) and 4C (G2) (*Figure 7—figure supplement 3O–P*). These results reaffirm our hypothesis that a cell's ability to respond to the target must be restricted to G2 in order for ATML1 to be predictive only in the G2 phase of the cell cycle.

We then asked whether our model could qualitatively reproduce the ATML1 dosage phenotypes we had observed with our genetic dosage series. We found that changing the basal ATML1 production rate was sufficient to gradually increase the total amount of the ATML1 in the modeled tissue, and accordingly, the fraction of giant cells in the sepal (*Figure 8*). These results, together with our dosage analysis, show that there is a positive relationship between graded ATML1 levels and the fraction of giant cells produced in the tissue (*Figures 2* and *8*).

Hence, our model shows that fast and relatively small stochastic fluctuations of ATML1 are sufficient to pattern giant and small cells in the sepal. ATML1 activates a downstream target, which if activated in G2, will induce endoreduplication. The dynamics of the ATML1-target network creates a soft ATML1 threshold during G2.



**Figure 6.** The dynamics of ATML1 fluctuations are independent of endoreduplication. (**A**) Raw images of *pATML1::mCitrine-ATML1* (white) from a live imaging series of a developing *lgo* mutant sepal. Images were taken every 8 hr for 64 hr. (**B**) Heat maps showing corresponding mCitrine-ATML1 concentrations (total fluorescence divided by area) at each time point from (**A**). (**C**) Genetic epistasis analysis between *lgo-2* mutant and *ATML1* overexpression line (*pPDF1::FLAG-ATML1*). Plants homozygous for both the *lgo* mutation and the overexpression transgene do not form giant cells, *Figure 6 continued on next page*

*Figure 6 continued*

demonstrating that LGO acts genetically downstream of ATML1 to promote endoreduplication. (D) Quantification of the average number of giant cells in four *pATML1::mCitrine-ATML1; atml1–3* sepals ($n_{cells}$ = 75, four sepals) compared to the number of giant cells predicted to form by applying the common threshold to ATML1 concentrations observed in *pATML1::mCitrine-ATML1; lgo* sepals ($n_{cells}$ = 59, three sepals). Error bars = standard error of mean. Approximately the same number of cells would be expected to become giant cells in *lgo* sepals as in wild type, except that they fail to endoreduplicate. A T-test performed between the two populations yielded a non-significant (ns) p-value of 0.9 (E) Traces of mCitrine-ATML1 normalized concentrations of cells that do not reach the inferred threshold in G2 of the cell cycle and are predicted to remain small ($n_{small}$ = 70). (F) Traces of mCitrine-ATML1 normalized concentrations of cells that reach the inferred threshold during G2 of the cell cycle and are predicted to become giant ($n_{giant}$ = 25). The trace ends when the cell is predicted to become giant. In (E–F) the dashed line represents the common normalized threshold derived from *pATML1::mCitrine-ATML1;atml1–3* flowers (*Figure 4—figure supplement 5*). (G–H) Single small cell lineages tracked through time (64 hr). Each nucleus image is outlined in a color associated with its ploidy: yellow = 2C, blue = 4C. The cell marked with X is lost from our tracking. (I–J) Tracked mCitrine-ATML1 concentration levels corresponding to the single cell lineages in (G–H). Cells that cross the mCitrine-ATML1 threshold fail to endoreduplicate and instead divide. A total of 149 lineages were analyzed (n = 495 cells). This flower is shown in *Video 8*. Two similar replicate flowers are shown in the *Figure 6—figure supplements 1* and *2*.

The following figure supplements are available for figure 6:

**Figure supplement 1.** Second flower showing that dynamic fluctuations of ATML1 are independent of endoreduplication.

**Figure supplement 2.** Third flower showing that dynamic fluctuations of ATML1 are independent of endoreduplication.

## Discussion

Here, we have identified a cell-autonomous fluctuation patterning mechanism for specifying cell fate in a multicellular system (*Figure 9*). During *Arabidopsis* sepal development, the pattern of giant cells and small cells in the epidermis is initiated through fluctuations of the transcription factor ATML1. Using live-imaging, quantitative image analyses and mathematical modeling, we have revealed that cells in which ATML1 levels surpass a soft threshold during the G2 phase of the cell cycle have a high probability of establishing giant cell identity and entering endoreduplication. A sepal epidermal cell is only competent to respond to ATML1 fluctuations during a window of time defined by G2 stage of the cell cycle.

Strikingly, our fluctuation-patterning model resembles Wolpert's French flag model in that each individual cell makes an autonomous fate decision based on the concentration of a key developmental regulator. Our model however deviates from the French flag model because it utilizes internal fluctuations instead of a diffusible morphogen to generate concentration differences. Concentration threshold-based patterning mechanisms have been traditionally viewed as being non-robust because they are sensitive to small perturbations in concentrations. Often additional mechanisms are needed

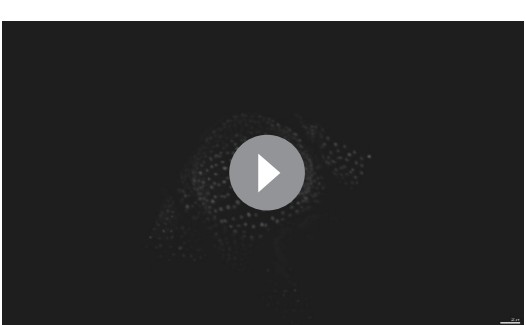

**Video 8.** A movie of a developing *pATML1::mCitrine-ATML1; lgo* sepal shown in *Figure 6*. The sepal primordium was live imaged every 8 hr throughout development.

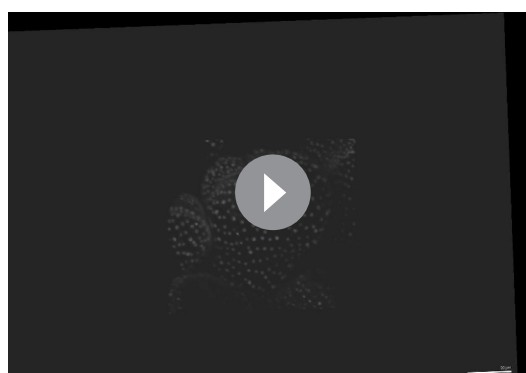

**Video 9.** A movie of a developing *pATML1::mCitrine-ATML1; lgo* sepal shown in *Figure 6—figure supplement 1*. The sepal primordium was live imaged every 8 hr throughout development.

to achieve robustness (**Eldar et al., 2002**, **2003**; **Kondo and Miura, 2010**). This sensitivity to small changes in concentration is consistent with our results in the sepal, where giant cell formation is highly responsive to changes in the basal production of ATML1. Interestingly however, in wild-type plants, the number of giant cells varies only slightly from sepal to sepal, falling within a small range (10-30). This indicates that these fluctuations together with a threshold must be tuned to ensure that the correct proportion of giant cells form on the sepal. Our data suggests that the cell cycle acts as a stabilizing factor to restrict giant cell fate decisions similarly to secondary mechanisms used in other biological systems.

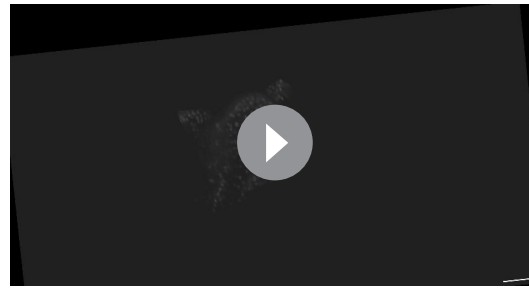

**Video 10.** A movie of a developing *pATML1::mCitrine-ATML1; lgo* sepal shown in **Figure 6—figure supplement 2**. The sepal primordium was live imaged every 8 hr throughout development.

A few recent studies have similarly demonstrated that the cell cycle provides a window of opportunity for making cell fate decisions. However, these studies suggest that G1 is the critical phase for specification. During G1, there is a growth factor-dependent restriction point, where a cell determines whether to enter quiescence (G0) or progress through the cell cycle. Cyclin/Cyclin Dependent Kinase (CDK) activity is normally reduced during the restriction point, providing a window for cells to receive extracellular signals necessary for cell fate decisions (**Blagosklonny and Pardee, 2002**; **Blagosklonny et al., 2002**). This has been nicely demonstrated in human embryonic stem cells, where a stem cell's ability to differentiate into an endodermal cell is dependent upon receiving TGF-$\beta$-Smad2/3 signals during this restriction point in early G1, when CyclinD levels are low (**Pauklin and Vallier, 2013**). In addition to transient Cyclin/CDK expression, some studies have found that cells extend their G1 phase immediately before differentiation. This may allow cell fate inducing factors to reach sufficient levels to induce differentiation (**Calegari et al., 2005**; **Collart et al., 2013**). How G1 lengthening occurs is still under debate. However, one recent study showed that increasing a cell's nuclear to cytoplasmic ratio dilutes the concentration of DNA replication factors which results in a prolonged G1 phase (**Collart et al., 2013**). Additionally, Singh et al. showed that chromatin changes associated with the M-G1 transition cause transcriptional leakiness of many prodifferentiation genes, which prime cells to respond to cellular differentiation signals (**Singh et al., 2013**).

We have found that a sepal epidermal cell's window to differentiate occurs not in G1 but in G2, suggesting a different manner of regulation than in G1-gated determination. For instance, cell fate decisions governed by the G1 phase of the cell cycle must often receive an extracellular signal to activate prodifferentiation genes instead of going into G0 quiescence. In contrast, our model suggests ATML1 fluctuations could be sufficient to pattern the sepal without a need for an extracellular signal. Alternatively, ATML1 could be priming the cell to receive a signal during the G2 phase of the cell cycle. We have previously reported that ACR4 (a transmembrane receptor kinase; **Gifford et al., 2003**, **2005**; **Watanabe et al., 2004**; **Roeder et al., 2012**) and DEK1 (a transmembrane calpain protease; **Liang et al., 2013**; **Lid et al., 2002**, **2005**; **Roeder et al., 2012**) act in the giant cell formation pathway, suggesting that intercellular signaling may assist in promoting giant cell fate decisions. An epistasis analysis between ACR4, DEK1 and ATML1 reveals that during giant cell formation, ACR4 acts upstream of ATML1 but that DEK1 acts downstream (**Figure 9—figure supplement 1**). These results are in opposition to what has been previously published about these genes during embryogenesis, where DEK1 acts upstream of ATML1 and ACR4 acts downstream (**Abe et al., 2003**; **Gifford et al., 2003**; **Johnson et al., 2008**; **San-Bento et al., 2014**; **Takada et al., 2013**; **Tanaka et al., 2002**). One possibility for these results is that ACR4 and DEK1 may act together with ATML1 in a feedback loop (**Galletti and Ingram, 2015**). As previously discussed (see Introduction), computational models propose that in tissues where no localized signals are present, stochastic fluctuations of transcriptional regulators create subtle differences between identical cells which initiate feedback loops including intercellular signaling to create the pattern (**Meyer and Roeder, 2014**). While our current model suggests that giant cell fate can be predicted through cell autonomous



**Figure 7.** A plausible stochastic model for giant cell patterning. (**A**) Schematic diagram of the computational model for giant cell patterning. Top panel shows the proposed ATML1 model network in which ATML1 can prevent cell division and instead drive entry into endoreduplication and giant cell specification. Middle panel shows a cartoon of the cell cycle timer time course. When the timer exceeds a first threshold level $\Theta_{C,S}$, cells enter into the G2 phase and increase their ploidy to 4C. When the timer reaches a second threshold level, $\Theta_{C,D}$, cells divide, unless their target levels have surpassed the threshold $\Theta_T$ sometime during G2 phase. Bottom panel shows a scatter plot cartoon illustrating how a 'hard threshold' in the target levels results in a 'soft threshold' in ATML1. We refer to a hard threshold when levels right above or below the threshold will result in two different outcomes. If the target perfectly followed the dynamics of ATML1, its upstream regulator, and obeyed a deterministic dynamics, all cells that cross the target threshold $\Theta_T$ would also cross a corresponding hard ATML1 threshold. Hence, a hard threshold in the target would be effectively encoded as a hard threshold on its upstream regulator ATML1. In contrast, in our model, the target has a finite degradation rate, and stochastic dynamics, so that it is not a perfect follower of ATML1 dynamics; thus, a hard threshold in target levels (vertical red dashed line) results in a soft threshold in ATML1 (horizontal red dashed line). A cell close to the ATML1 soft threshold may or may not pass the target threshold and endoreduplicate to become a giant cell. Dots in the

*Figure 7 continued on next page*

*Figure 7 continued*

bottom panel is a cartoon of the ATML1 maxima of simulated cell lineages, with red dots indicating cells that become giant, while blue dots represent mitotically dividing small cells. (B) Simulation snapshots of the *in silico* growing sepal showing (top) ATML1 concentrations and (bottom) cell ploidies (*Video 11*). (C) Time courses of ATML1 (left) and its target (right) for a cell committing to the giant fate (top) and a small dividing cell (bottom). Colors of the time traces represent the cell ploidy. Color code for the ploidies is the same as in panel B. Red dashed lines represent the predicted soft ATML1 threshold $\Theta_A^*$, and the $\Theta_T$ hard threshold imposed in the target (Materials and methods). (D) Histogram at a final simulation time point showing ATML1 concentration levels. (E) Boxplot showing the percentage of cell ploidies in a simulated tissue for five simulations with different random initial ATML1, target and timer levels. (F–G) ROC analysis of the ATML1 concentration maxima for the simulated lineages at (F) 2C and (G) 4C, showing that the ATML1 maximal levels at 2C is not predictive, in agreement with experimental data (*Figure 4F–I*; *Figure 4—figure supplements 1–3F–I*). Parameter values are described in *Table 1*.

The following figure supplements are available for figure 7:

**Figure supplement 1.** Simulation results showing different stochastic time courses.

**Figure supplement 2.** Stochastic fluctuations are essential for generating the giant cell patterning.

**Figure supplement 3.** Classification analysis of the simulated data shows that a weak feedback or no feedback in ATML1 reproduces the experimental observations.

**Figure supplement 4.** Theoretical and experimental study of the ATML1 auto-induction strength.

mechanisms, it will be interesting to see if ACR4 and DEK1 act to help establish or maintain giant cell fate or to propagate giant cell patterning in the developing sepal.

To facilitate the entry into endoreduplication, ATML1 may need to activate a downstream target that only functions during G2 phase of the cell cycle. One possible ATML1 target is the *Siamese*-related CDK inhibitor LGO. LGO acts genetically downstream of ATML1 in the giant cell pathway to promote endoreduplication once giant cell identity is acquired (*Figure 6C*; *Roeder et al., 2012*). It is not yet exactly understood how CDK inhibitors like LGO function in promoting endoreduplication because some evidence suggests that they interact with cyclin-CDK complexes during both G1-S and G2-M transitions, while other studies suggest specificity for G2-M (*Boudolf et al., 2009*; *Churchman et al., 2006*; *Kumar et al., 2015*; *Van Leene et al., 2010*). It is hypothesized that SIA-MESE and LGO control the entry into endoreduplication by inhibiting G2-M transitions (*Kalve et al., 2014*; *Roodbarkelari et al., 2010*; *Van Leene et al., 2010*). It will be interesting to test whether the G2 responsiveness of ATML1 arises due to direct or indirect regulation of LGO.

There are a few examples that support the idea that G2 can be important for post-mitotic cell differentiation. For instance in *Drosophila*, changes in protein levels of the homeobox transcription factor Pax6 during the G2-M transition will cause neurogenic progenitor cells to specify into different types of post-mitotic neurons (*Hsieh and Yang, 2009*). Although Pax6 behaves similarly to ATML1 through controlling cell fate in a dosage dependent manner, *Pax6* expression remains relatively constant in neurogenic progenitor cells until the G2/M phase. This indicates that Pax6 does not undergo random fluctuations like ATML1, but is likely regulated by an upstream factor. Other examples of G2 mediated cell fate decisions include the development of secondary vulval precursor cells, where precursor cells

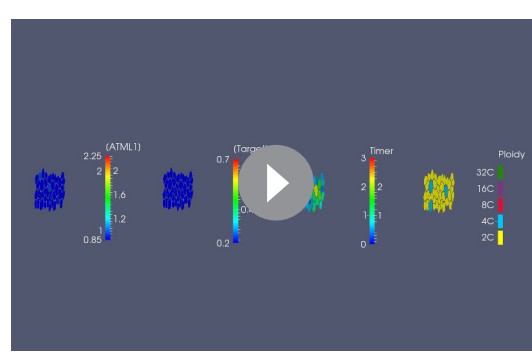

**Video 11.** Simulation results showing ATML1, target, timer levels and cell ploidies throughout time in a growing tissue. Cells that cannot divide, increase their ploidy, becoming giant cells. The time resolution of the displayed movie (0.5) is lower than the actual simulation time step (0.1), so fluctuations in ATML1 and in the target may be missed. Color scales in the ATML1 and target variables have been truncated for the sake of better visualizing the fluctuations. Parameter values are described in *Table 1*.

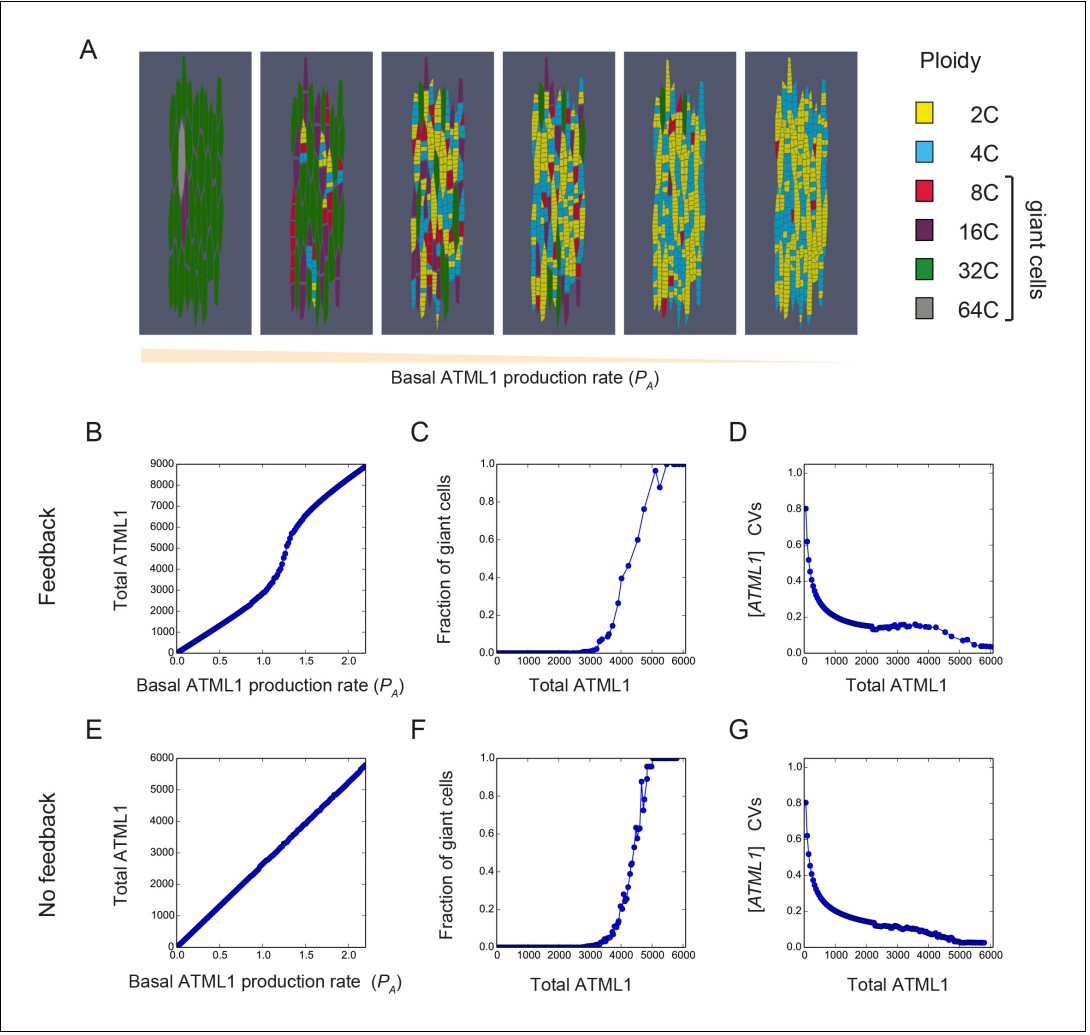

**Figure 8.** The model recapitulates *ATML1* dosage dependency. (**A**) Snapshots showing the resulting patterns of giant cells (8C, 16C, 32C and 64C cells) and small cells (2C and 4C cells) at the final time point of the simulations when the basal ATML1 production rate is modified. Values chosen for the ATML1 basal production rate from the parameter exploration shown in panels B-G are, from left to right: $P_A$ = 1.58, $P_A$ = 1.25, $P_A$ = 1.17, $P_A$ = 1.14, $P_A$ = 1.01 and $P_A$ = 0.99. (**B–G**) Simulation results for different basal ATML1 production rates for (**B–D**) a model with a weak auto-induction ATML1 feedback loop ($V_A$ = 1.25) and for (**E–G**) a model with no feedback ($V_A$ = 0). (**B** and **E**) Total amount of ATML1 in the tissue. The total ATML1 amount is the sum of the area of each cell multiplied by the ATML1 concentration in that cell. The feedback drives a sharper increase of ATML1 amount for a certain range of basal ATML1 production rates. (**C** and **F**) Fraction of giant cells (8C, 16C, 32C and 64C cells) in the tissue with respect to the total amount of ATML1. The gradual increase of the fraction of cells with respect to the total ATML1 amount in the tissue is qualitatively consistent with the different phenotypes shown in *Figure 2*. The model with feedback has a slightly more gradual increase in fraction of giant cells with respect to the total amount of ATML1. (**D** and **G**) CVs of the ATML1 concentrations in the tissue. In the cases of having a weak feedback or not having a feedback, there is a plateau of CV values for intermediate ATML1 total amounts in the tissue. Stronger feedback levels will lead to non-monotonic CVs with respect to the total amount of ATML1 (see *Figure 7—figure supplement 2B*). Other parameter values are described in *Table 1*.

require high levels of LIN-12 mediated signaling during G2 to commit to secondary cell fates (*Ambros, 1999*), and *Drosophila* mechanosensory precursor cells, where cells enter a temporary quiescence in G2 to provide a small window for proneural determinant gene products to accumulate (*Nègre et al., 2003*). Although both systems use G2 as a window to initiate cell fate decisions, neither has been reported to experience fluctuations similar to ATML1.

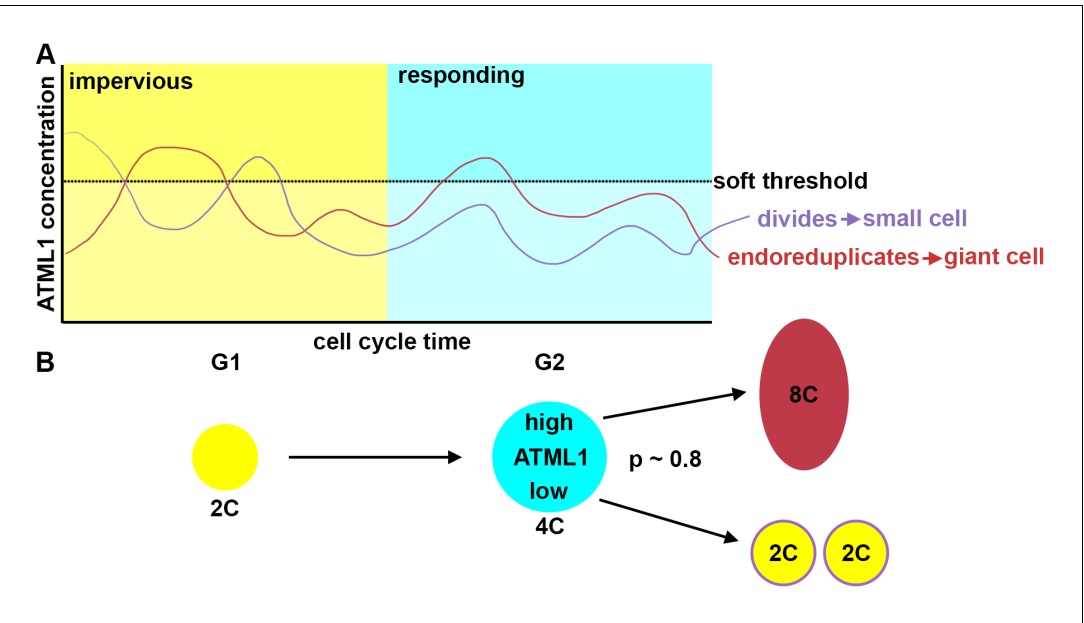

**Figure 9.** Fluctuations of ATML1 around a soft threshold pattern giant cells and small cells in the sepal. ATML1 fluctuates in every young sepal epidermal cell. However, cells only respond to high levels of ATML1 during G2 phase of the cell cycle. (**A**) Schematic showing that in G1, cells are impervious to high concentrations of ATML1. In G2, cells can respond to ATML1 to become a giant cell if levels surpass a soft threshold. If a cell does not receive a high enough level then the cell will divide. (**B**) Schematic demonstrating a cell progression from 2C (G1 phase of the cell cycle) to 4C (G2 stage of the cell cycle). The cell will then either become an 8C cell, if it receives a high level of ATML1, or to divide to make two 2C cells if ATML1 levels are low. In the G2 phase, our inferred mCitrine-ATLML1 threshold level is about 80% accurate in predicting giant cells versus small identity correctly.

The following figure supplement is available for figure 9:

**Figure supplement 1.** ACR4 and DEK1 act in the giant cell patterning pathway.

Our theoretical model has shown that dynamic stochastic fluctuations in protein expression levels can provide a mechanism for singling out cells in the developing sepal to adopt the giant cell fate. It would be interesting to examine whether other sources of noise can shape such fluctuations and contribute to the process of giant cell fate commitment. In our giant cell patterning model, a hard threshold in the downstream target produces a soft threshold in the upstream regulator (i.e. ATML1). A soft but still reliable threshold can emerge when a target follows the dynamics of its upstream regulator. Indeed, our experimental data shows that the ATML1 threshold is soft, but robust across different plants.

We have described a cell-autonomous fluctuation-driven patterning mechanism, where fluctuations of the transcription factor ATML1 must reach a concentration threshold during the G2 stage of the cell cycle to regulate cell fate decisions. This overall demonstrates that stochastic processes can be important for creating spatial patterns necessary for reproducible tissue development.

# Materials and methods

## Plant accessions
Columbia (Col) plants were used as the wild-type accession for all genotypes except *pSEC24A::H2B-GFP* which was in Landsberg *erecta* (**Qu et al., 2014**).

*atml1–3* (SALK_033408); exhibits a lack of giant cell phenotype. The *atml1–3* mutation is a dosage dependent mutation that contains a T-DNA insertion in the homeodomain. The *atml1–3* mutation can be PCR genotyped by amplifying with oAR272 (CAGGCAGAAGAAAATCGAGAT), oAR273 (GAAACCAGTGTGGCTATTGTT) and LBb1 (GCGTGGACCGCTTGCTGCAACT).

*lgo-2* (SALK_039905); exhibits a lack of giant cell phenotype. The *lgo-2* mutation is a recessive mutation, containing a T-DNA insertion. The *lgo-2* mutation can be PCR genotyped by amplifying with oAR284 (CTTCCCTCTCACTTCTCCAA), oAR285 (CCGAACACCAACAGATAATT), and JMLB2 (TTGGGTGATGGTTCACGTAGTGGG) (*Roeder et al., 2010*).

*dek1–4* plants do not form giant cells. The *dek1–4* mutation can be PCR genotyped by amplifying with oAR448 (TGTTGGTGGAACAGACTATGTGAATTCA) and oAR449 (TGAAGACTGAAAGGA-CAAAAGGTGC) with a 60°C annealing temperature followed by a 4 hr product digest using BsaAI.

*acr4–24* plants have a severe reduction in the number of giant cells that form. The *acr4–24* mutation can be PCR genotyped by amplifying with oAR302 (ATAGAAGTCCCTGTGAGAACTGCG) and oAR303 (TATGATCATAGTGCGGTCTGTTGG) with a 60°C annealing temperature followed by a 4 hr product digest using HhaI.

*pAP2::AP2-2XYpet* plants were provided by Jeff Long (*Wollmann et al., 2010*).

*pVIP1::VIP-mCitrine* plants were provided by the ABRC (CS36991) (*Tian et al., 2004*).

ATML1 estradiol inducible lines were provided by Shinobu Takada (*proRPS5A-ATML1/pER8* and *proATML1-nls-3xGFP*) and Keiko Torii (*pKMP151 line #134*) (*Peterson et al., 2013*; *Takada et al., 2013*).

All plants used for this analysis were grown in Percival growth chambers with 24 hr light conditions at 22°C to minimize any diurnal effect on plants.

## Accession numbers

*ATML1*, AT4G21750; giant cell enhancer trap marker, YJ158; small cell enhancer trap marker, CS70134; *LGO*, AT3G10525; *atml1–3*, CS68906, SALK_033408; *lgo-2*, CS69160, SALK_039905; *pPDF1::GFP-ATML1*, GIL91–4; *pPDF1::FLAG-ATML1*, GIL90–5; *SEC24*, AT3G07100; *pVIP1::VIP-mCitrine,* CS36991; *CER5*, AT1G51500; *FDH*, AT2G26250; *PDF2*, AT4G04890; *PDF1*, AT2G42840.

## Genetic crosses

To create genetically altered lines of ATML1 for our dosage series, we first crossed *PDF1::FLAG-ATML1* plants, which exhibit an all ectopic giant cell phenotype to Columbia plants, resulting in F1 plants that were hemizygous for the *PDF1::FLAG-ATML1* transgene (*PDF1::FLAG-ATML1/+*). To lower amounts of ectopic ATML1 even further, *PDF1::FLAG-ATML1/+* plants were crossed into the *atml1–3* mutant background. Using genetic segregation and PCR genotyping, plants containing the *PDF1::FLAG-ATML1* transgene in an *atml1–3* mutant background were recovered and analyzed (*PDF1::FLAG-ATML1/+; atm1–3/atml1–3*). Next, to look at the effects of *atml1–3* heterozygotes, the Columbia plants were crossed with *atml1–3* mutants. The resulting F1 plants were analyzed.

To assess whether ectopic sepal giant cells from *PDF1::FLAG-ATML1* plants confer giant cell identity, *PDF1::FLAG-ATML1* plants were crossed with plants expressing the giant and small cell marker (PAR111 and CS70134; *Roeder et al., 2012*). Plants homozygous for all three transgenes were analyzed.

To look at the effects of ATML1 in flowers that lack giant cells, *pPDF1::FLAG-ATML1* plants were crossed with giant cell patterning mutants *lgo-2*, *acr4–24*, and *dek1–4*. Genotyping PCR was used to identify plants homozygous for *pPDF1::FLAG-ATML1* and either *lgo-2*, *acr4–24*, or *dek1–4*.

To see how *pATML1::mCitrine-ATML1* behaved in *lgo-2* mutants, *pATML1::mCitrine-ATML1* plants were crossed into *lgo-2* mutants. Genetic segregation analysis and confocal microscopy was used to find *pATML1::mCitrine-ATML1; lgo-2* plants.

## Microscopy

Scanning electron microscopy was performed as previously described (*Roeder et al., 2010*). Briefly, Stage 14 flowers were fixed in an FAA solution (50% ethanol, 5% acetic acid, and 3.7% formaldehyde) for 4 hr and dehydrated using an ethanol series. Flowers were critical point dried and sepals were dissected. Sepals then were sputter-coated with platinum palladium and imaged using a LEICA 440 scanning electron microscope.

Analysis of the giant and small cell enhancer fluorescent reporters was performed as previously described in (*Roeder et al., 2012*). Stage 12 medial abaxial sepals were stained with Propidium Iodide (PI) and imaged with a Zeiss 710 laser scanning confocal microscope. The small cell marker was excited with a 488 nm laser and emission was collected with a 493–516 filter whereas the giant

cell enhancer was excited with a 514 nm laser and emission was collected with a 519–565 filter. PI emission was collected with a 599–651 filter. Images were taken with a 10x objective.

*pSEC24A::H2B-GFP* was imaged using a Zeiss 710 laser scanning confocal microscope. The GFP marker was excited with a 488 nm laser and collected with a 493–548 filter. Nuclear fluorescence was then calculated using our quantification pipeline.

*pVIP1::VIP1-mCitrine* was imaged using a Zeiss 710 laser scanning confocal microscope. The mCitrine marker was excited with a 514 nm laser and collected with a 519–564 filter. VIP1 is a bZIP transcripton factor that is cytoplasmically localized under stable conditions but will become nuclear localized upon hypoosmotic treatment (*Tsugama et al., 2016*). To nuclear localize VIP1, VIP1-mCitrine inflorescences were submerged in a hypoosmotic solution (H2O and 0.001% triton-X) for approximately 10 min prior to confocal imaging. Nuclear fluorescence was then calculated using our quantification pipeline.

*pAP2::AP2-2XYpet* was imaged using a Zeiss 710 laser scanning confocal microscope. The 2XYpet marker was excited with a 514 nm laser and collected with a 519–564 filter. Nuclear fluorescence was then calculated using our quantification pipeline.

Live imaging of each fluorescent reporter line in developing sepals was performed as previously described (*Roeder et al. 2010*), except for the experimental setup. Transgenic plants including pHM44 *pATML1::mCitrine-ATML1* (ex. 514 nm at 2%, em. 519–564 nm), *lgo*;pHM44 *pATML1::mCitrine-ATML1* (ex. 514 nm at 2–2.2%, em. 519–564 nm) or GIL91–4 *pPDF1*::GFP-ATML1 (ex. 488 nm at 1–1.5%, em. 493–598) were imaged either every 8 hr or every hour using a Zeiss 710 laser scanning confocal microscope with a 20x water-immersion objective (numerical aperture = 1.0). Before imaging, plant inflorescences were dissected down to early stage flowers and meristems and then taped onto slides. Dissected inflorescences were then stained with PI and mounted with a cover slip and imaged. Inflorescences were unmounted, dried, and plants were placed upright in the growth chamber for 8 hr before remounting and imaging. The resulting images were 3D cropped with ImageJ (*Schindelin et al., 2012*; *Schneider et al., 2012*) to remove neighboring flowers. mCitrine-ATML1 fluorescence was quantified in each nucleus throughout the live imaging series with our pipeline (see below).

## DNA and cell size quantification

Flow cytometry was conducted as previously done in (*Roeder et al., 2010*) using an Accuri C6 flow cytometer. 50–100 stage 12 sepals were dissected from transgenic plants containing epidermal GFP-tagged nuclei (pAR180 pML1::H2B-mGFP). Nuclei were stained with PI and gated as described previously (*Roeder et al., 2010*) to isolate epidermal nuclei (GFP positive) from internal tissue nuclei (GFP negative). PI fluorescence histograms showed the relative DNA content of each population analyzed.

Ploidy and nuclear area were quantified from DAPI stained sepals as previously described (*Roeder et al., 2010*) and imaged with a Zeiss 700. DAPI was excited with a 405 nm laser and emission collected with a 410–584 nm filter. Images were cropped in ImageJ and quantified using ImageJ or our quantification pipeline.

Cell size analysis was performed by imaging pAR169 (*pML1::mCitrine-RCI2A*) sepals with a Zeiss 710 confocal microscope. mCitrine was excited with a 514 nm laser and emission was collected with a 519–621 filter. Imaged sepals were semi-automated image processed using a MATLAB module, which has been previously published (*Cunha et al., 2010*; *Roeder et al., 2010*) to determine cell area.

## Transgenes

To create pHM44 (p*ATML1::mCitrine-ATML1*), a 6160 bp fragment upstream of the ATML1 protein coding region was PCR amplified using oHM23 (ACC GAC AAT GTA TGAA TGT ACT CT) and oHM24 (cgg tac cgg cgc gcc GAT GAT GAT GGA TGC CTA TCA ATT T) and cloned into a pGEM-T Easy vector to create pHM20. Additionally, a 992 bp region downstream of the ATML1 protein coding region was PCR amplified using oHM25 (cgg tacc TCG ATG TTT TCG GGT AAG CTT TTT) and oHM26 (TTT GAT GAC TTG GTC TCC ATA ATT TC) and cloned into pGEM-T easy to create pHM21. pHM21 was cut with SacII and KpnI and cloned into pHM20 to make pHM22. A gateway cassette from pXQ (AscI-GW-KpnI in pGEM-T easy) was cut with AscI and KpnI and cloned into pHM22 to make pHM23

(*Qu et al., 2014*). Then, pHM23 was cut with NotI and cloned into the pART27 binary vector to make pHM43 (*pATML1::GW:ATML1 3'UTR*). Next, mCitrine was PCR amplified using oHM42 (CAC CAA AAT GGT GAG CAA GGG CGA GGA GCT G) and oHM39 (atA CTA GTG GCC GCT GCC GCA GCG GCA GCC GCA GCT GCT CCG GAC TTG TAC) and cloned into pENTR/D-TOPO vector to make pHM30. ATML1 was PCR amplified using oHM40 (tcg gcg cgc cCA CCC TTT TAG GCT CCG TCG CAG GCC AGA GCG GCT) and oHM41 (cca ctag tAT GTA TCA TCC AAA CAT GTT CGA ATC TCA TC) and cloned into pGEM-T easy to make pHM28. ATML1 was cut using SpeI and AscI and cloned into pHM28 to make pHM25 (pENTR *mCitrine-ATML1*). LR reaction between pHM25 and pHM43 to make pHM44 (p*ATML1::mCitrine-ATML1*). The *atml1–3* rescue line was generated by transforming the pHM44 p*ATML1::mCitrine-ATML1* transgene into *atml1–3* mutants using *Agrobacterium*-mediated floral dipping methods (*Clough and Bent, 1998*). We recovered lines with varying numbers of giant cells, presumably due to varying levels of transgene expression. From the lines recovered, two produced the wild-type number of giant cells, rescuing the mutant phenotype. Both of these lines showed differing levels of ATML1 among cells. Therefore, we characterized one of them.

## ATML1 estradiol induction

To test whether ATML1 acts in a feedback loop, inflorescences of ATML1-estradiol inducible plants (*proRPS5A-ATML1/pER8* and *proATML1-nls-3xGFP* line #7 provided by *Takada et al., 2013*) were cultured in apex culturing media (1/2x MS, 1% sucrose, 0.5 g/L MES, pH 5.7, 0.8% agar; *Hamant et al., 2013*) containing either 0.1 μM, 1 μM, or 10 μM estradiol or a mock solution (ethanol equivalent to the solvent of estradiol). Tissue was then collected 48 hr later and prepared for qPCR. Three or five biological replicates were analyzed for each treatment (estradiol and mock).

To test whether inducing ATML1 could increase the number of giant cells that form on the sepal, we dipped inflorescences expressing *proRPS5A-ATML1/pER8* and *proATML1-nls-3xGFP* provided by Shinobu Takada (*Takada et al., 2013*) in 10 μM estradadiol (with 0.01% silwet) for three consecutive days and then examined seven sepals (stage 8–10) five days later and compared them to untreated sepals at equivalent developmental stages.

## Quantitative PCR

To perform qPCR, 3–4 inflorescences were collected per sample and total RNA was extracted using RNeasy Plant Mini Kit (Qiagen, Venlo, Netherlands). Next, 1 microgram of total RNA was DNAse treated with amplification grade DNAse I (Invitrogen, Carlsbad, USA) and reverse transcribed using Superscript II reverse transcriptase (Invitrogen) with oligo dT primers. Real-time PCR was performed using 480 SYBR Green I Master (Roche, Indianapolis, IN) on a Roche *LightCycler* 480 system. At least three biological replicates were analyzed per genotype and ROC1 (AT4G38740) was used as a reference gene to normalize gene expression. Furthermore, three technical replicates were used to ensure the validity of each biological replicate.

qPCR primers:

- oHM58: GAG CTA GAG TCG TTC TTC AAG G – qPCR forward primer for *ATML1* (flanks *atml1–3* insertion)
- oHM62: GTT CTC GTG CCT CTC ATG TTG TG – qPCR reverse primer for *ATML1* (flanks *atml1–3* insertion)
- atml1-ATGF: GGA TAT ACA GGC AGA AGA AAA TCG AG – qPCR forward primer for endogenous *ATML1* 5'UTR (upstream of start site)
- oAR715: CGC TGA AGC TAG TCG ACT CTA – qPCR forward primer for induced *ATML1* 5' UTR (specific to UTR of induction construct, not found in genome)
- oAR716: TTC TCC ATG GTG ACT TCT GCG – qPCR reverse primer for *ATML1* (just downstream of the start codon in both the endogenous and induced transcripts).
- CER5-qPCR1: AGG AAT ATC GCT CGA GAT GG – qPCR forward primer for *CER5* (*Takada et al., 2013*)
- CER5-qPCR2: TGT CTC CCG AAT CCT TTG AG – qPCR reverse primer for *CER5* (*Takada et al., 2013*)
- FDH-qPCR1: TTC CGC CAC CGC AAA AAC CAA TG – qPCR forward primer for *FDH* (*Takada et al., 2013*)
- FDH-qPCR2: TGC CGC GTG GAA GCA AAA ATG C – qPCR reverse primer for *FDH* (*Takada et al., 2013*)

- PDF2-qPCR1: TCC GCG AAG AGA TTG ATA GG – qPCR forward primer for *PDF2* (*Takada et al., 2013*)
- PDF2-qPCR2: AGA TCA AGC GAA CGA GAA GG – qPCR reverse primer for *PDF2* (*Takada et al., 2013*)
- PDF1-qPCR1: TGA GTT TTG CCG TTT GGG CTC TC – qPCR forward primer for *PDF1* (*Takada et al., 2013*)
- PDF1-qPCR2: TGT GGA GTT GGC GTG TGT GAT GG – qPCR reverse primer for *PDF1* (*Takada et al., 2013*)
- Cyclo-F: CGA TAA GAC TCC CAG GAC TGC CGA – qPCR reference forward primer for *ROC1*
- Cyclo-R: TCG GCT TTC AGA TGA TGA TC CAA CC – qPCR reference forward primer for *ROC1*

## Image analysis and quantification pipeline

In order to accurately quantify mCitrine-ATML1 levels at the single cell level and track individual cells during sepal growth, we developed an integrated image analysis pipeline incorporating modules from different available sources.

## Preprocessing and segmentation

Raw fluorescence intensity images were denoised using the PureDenoise ImageJ plugin (*Blu and Luisier, 2007*; *Luisier et al., 2009*, *2010*), optimized for the mixed Poisson-Gaussian noise that typically affects fluorescence microscopy images (parameters: frames = 4; cycle spins = 3). Denoised images were imported into MorphoGraphX (*Barbier de Reuille et al., 2015*) in order to produce binary masks for individual sepal nuclei while simultaneously removing non–relevant meristematic and border cell nuclei (parameters: brighten/darken: 1–4; Gaussian Blur: 0.3–1; Binarize: 5000–8000). Since different genotypes show different proportions of giant and small cells, and segmentation parameters are globally applied to the whole tissue, slight adjustments were made for each genotype in order to fit the binary masks as well as possible to all nuclei across all genotypes. For each individual time course, parameter values were kept constant for all time points. Binary mask images were used as input for the final nuclear segmentation, performed with the Costanza (COnfocal STack ANalyZer Application ImageJ plugin (http://www.plant-image-analysis.org/software/costanza). Costanza performs segmentation following the steepest descent algorithm, providing high-resolution three-dimensional segmentation of each individual sepal nucleus.

## Cell tracking

Denoised images were processed using the FeatureJ ImageJ plugin (http://imagej.net/FeatureJ) for edge detection by applying the Canny method (parameters: gradient-magnitude image smoothing scale = 0.25). To track the cell nuclei between two successive nuclei segmentations, $N_t$ and $N_{t+\Delta t}$ (where $\Delta t$ corresponds to the time interval between two consecutive acquisitions), the block matching framework (*Michelin et al., 2016*, *Commowick et al., 2008*; *Ourselin et al., 2000*) was used to non-linearly register the corresponding denoised images, $I_t$ and $I_{t+\Delta t,}$ (floating and reference images respectively). The registered floating image and the reference image were merged with different colors into a double channel image in ImageJ (*Box 1*, 3D projection of the merged image, red: reference image, green: registered floating image). This allowed a visual inspection of registration quality. The non-linear transformation computed by block matching, $T_{It \leftarrow It+\Delta t}$, was then applied to $N_t$ (i.e. $N_t \circ (T_{It \leftarrow It+\Delta t})$. Using ALT (*Fernandez et al., 2010*) we computed optimal cell-cell pairing between $N_t$ and $N_t \circ nT_{It \leftarrow It+\Delta t}$. Given the spatial complexity of the tissue and the large time interval between consecutive images ($\Delta t$=8 hr), registration was not always successful for all nuclei. Incorrectly tracked nuclei were manually corrected using the MorphoGraphX parent labels tools, making use of the ALT-generated optimal pairing tables, describing the mother/daughter relations between time points.

## Quantification and analysis

A set of MATLAB (The MathWorks, Inc., Natick, Massachusetts, United States) functions and scripts was developed to quantify signal intensity, as well as size and shape properties of individual nuclei from sets of confocal microscopy images processed as described above (*Source code 1*). We did not use a secondary nuclear marker to detect nuclear size because mCitrine-ATML1 levels are low

and may experience bleed through from a nuclear marker in a different channel. Additionally, given imaging artifacts observed when using three-dimensional images, which include extension of nuclei in the Z-axis, we chose to perform quantification in two-dimensional images, in order to maximize result accuracy. Two-dimensional nuclei were obtained by scanning, for each nucleus, through each individual Z slice of the Costanza-segmented images and selecting the slice with the largest area, where segmentation is most accurate. For all 2D nuclei, shape parameters such as eccentricity were quantified using the *regionprops* function in the MATLAB Image Processing Toolbox, which was also used to quantify areas. For each 2D nucleus, absolute fluorescence intensity was quantified by summing intensity of all pixels in the respective region of the raw intensity image. Both absolute intensity and area were corrected for possible magnification changes during the time course by taking into account pixel sizes, and concentrations were calculated based on the corrected absolute intensity and area values. ATML1 concentration, area and eccentricity plots for all cells in the time course were generated with custom functions that make use of the corrected parental correspondence information. From the complete set of tracked lineages, we selected for lineages that exhibited high quality segmentation and tracking data that allowed us to follow a given cell either until the last point of the time course, or a until fate became apparent (division or endoreduplication; see *Supplemental file 1* for examples).

## Receiver operator characteristics (ROC) analysis

Calculations were performed using the *perfcurve* function of the Statistics and Machine Learning Toolbox in MATLAB (*Source code 1*). Classes were defined based on their final identity (small or giant) and cell cycle stage (2C for G1 or 4C for G2). After ploidy was assigned, we identified peaks in mCitrine-ATML1 concentration at G1 and G2 stages of the cell cycle for ROC analysis. Individual cell lineages were included in the analysis only if a cell passed through both the G1 and G2 stages of the cell cycle before entry into either mitosis or endoreduplication or was first detected in G2 and remained in G2 for more than two consecutive time points before entry into mitosis or endoreduplication. For these lineages we used the highest concentration level of mCitrine-ATML1 during both the G1 and G2 stages of the cell cycle before entry into either mitosis (small cell) or endoreduplication (giant cell).

For each sepal, the ATML1 concentration value that maximized the difference between true positive rate (TPR) and false positive rate (FPR) when classifying small versus giant cells, was taken as the threshold ATML1 concentration required for triggering giant fate decision in individual cells. For sepals where such a threshold could not be inferred, whether due to the absence of sufficient numbers of dividing (*pPDF1::GFP-ATML1* flowers) or endoreduplicating cells (*lgo-2* mutant flowers), a common threshold was inferred from the wild-type *pATML1::mCitrine-ATML1* flowers (*Figure 4—figure supplement 5*). For each of the four analyzed sepals, ATML1 concentrations were normalized by dividing by the mean concentration for all nuclei, over the entire time course. This mean normalization had the objective of taking into account systematic differences between time courses due to experimental variation. After normalization, ATML1 concentration peak selection, ROC analysis and concentration threshold inference were performed as described above. The final common ATML1 concentration threshold was defined as the mean of the four individual thresholds.

In the *lgo-2* mutant sepals, giant cell fate prediction was performed by comparing the normalized ATML1 concentrations for each lineage (calculated using the mean concentration for all nuclei, over the entire time course) with the previously inferred common threshold. A lineage was considered to be a giant cell lineage if the ATML1 concentration of a given nucleus surpassed the common threshold concentration during 4C at any point of the time course. If this event never occurred, the lineage was considered to correspond to a small cell lineage.

## Theoretical model

We implemented a stochastic computational model for ATML1 mediated giant cell fate decisions in a 2D idealized growing tissue. The model has a core simplified ATML1 regulatory network that can prevent cell division, driving cell endoreduplication. We modeled ATML1 cell concentration dynamics as a basally produced protein that self-activates and is linearly degraded (*Frigola et al., 2012*; *Weber and Buceta, 2013*). ATML1 expression activates a downstream target, which can prevent cell division when expression passes a threshold. The deterministic expression for the dynamics of

ATML1 and its downstream target concentrations in cell $i$, whose variables are $[ATML1]_i$ and $[Target]_i$ respectively, reads

$$\frac{d[ATML1]_i}{dt} = P_A + \frac{V_A[ATML1]_i^{n_A}}{K_A^{n_A} + [ATML1]_i^{n_A}} - G_A[ATML1]_i \tag{1}$$

$$\frac{d[Target]_i}{dt} = \frac{V_T[ATML1]_i^{n_T}}{K_T^{n_T} + [ATML1]_i^{n_T}} - G_T[Target]_i, \tag{2}$$

where $P_A$ is a basal ATML1 production rate, $V_X$ is the maximal ATML1-dependent production rate for the $X$ (either ATML1 or Target concentration) variable, $K_X$ is the ATML1 concentration at which the ATML1-dependent production rate has its half-maximal value, $n_X$ is the Hill coefficient and $G_X$ is the linear degradation rate for the $X$ variable. For simplicity, we will refer to $V_A$ as the ATML1 auto-induction rate, so no feedback is considered when $V_A = 0$.

A cell is defined by a set of vertices in 2D and we set the tissue to grow exponentially and aniso-tropically by moving vertices outwards from the center of mass of the tissue. Hence, all cells grow anisotropically, and they divide according to a timer variable present in each cell. We implemented dilution of ATML1 and its target variables due to growth. During sepal development, nuclear and cell area of epidermal cells are correlated (*Figure 4—figure supplement 4*). We used cell area growth to implement the dilution effect into the ATML1 and target variables. The timer linearly increases with time and is reset when it reaches a specific threshold (*Figure 7A*). Hence, its equation reads

$$\frac{dTimer_i}{dt} = P_C, \tag{3}$$

where $Timer_i$ is a variable in cell $i$, and $P_C$ is the basal timer production rate. The timer resetting was performed at each time step according to the following equation:

$$Timer_i(t) \rightarrow \begin{cases} U_i & if \ Timer_i(t) \geq \Theta_{C,D} \\ Timer_i(t) & otherwise \end{cases}, \tag{4}$$

where $U_i$ is a uniform randomly distributed number in the interval [0, 0.5] and $\Theta_{C,D}$ is a cell division threshold for the timer.

Cell ploidy was modeled as a discrete variable dependent on the timer and cell division, which also depends on the ATML1 network. Specifically, cell ploidy increases from 2C to 4C when the timer reaches a threshold $\Theta_{C,S}$, which represents S phase, and decreases again to 2C if the cell divides. Cell division occurs at the 4C stage, when the timer reaches a second threshold $\Theta_{C,D}$, unless cells have reached [*Target*] levels higher than a specific threshold $\Theta_T$ during the 4C stage. In that case, endoreduplication occurs, and cells reset their timer when they reach the $\Theta_{C,D}$ threshold, but keeping its ploidy to 4C. We imposed that 4C cells having endoreduplicated once cannot undergo cell division anymore. As a consequence, these cells will increase their ploidy every time they pass the timer threshold $\Theta_{C,S}$ representing entry into S phase.

Our experimental data shows that the nuclear area scales with the DNA content and ploidy in the cell (*Box 2*; *Box 2—Figure 1*; *Box 2—Videos 1–2*). Previous data in tomato has shown that expression levels positively correlate with cell ploidy (*Bourdon et al., 2012*), so one could also assume there is a linear correlation between ploidy and expression levels. Because of these two assumptions, the production rates of the ATML1 and Target concentration variables become independent of the cell ploidy. For the sake of simplicity, production rates remain constant throughout cell cycles.

Dynamic stochasticity was introduced in the ATML1, Target and Timer variables by extending its deterministic dynamics to its Langevin form (*Adalsteinsson et al., 2004*; *Gillespie, 2000*). In particular, for every ATML1, Target and Timer variable $X$ in cell $i$, the resulting stochastic equations would read

$$\frac{dX_i}{dt} = F_{Xi}^+ - F_{Xi}^- + \sqrt{\frac{F_{Xi}^+ + F_{Xi}^-}{2\varepsilon_i(t)}} \eta_{Xi}(t), \tag{5}$$

where $F_{xi}^+$ and $F_{xi}^-$ are positive functions that represent the birth and death processes for the

species $X$ in cell $i$. Hence, we take into account stochasticity coming from production and degradation of the modeled species. $\varepsilon_i(t)$ is a normalized cell area; we assume $\varepsilon_i(t)=E_0E_i(t)$, where $E_0$ is an effective cell area used to normalize noise, and $E_i(t)$ is the area of cell $i$ in arbitrary units. $\eta_{Xi}$ is a random Gaussian variable with zero mean that fulfills $\langle \eta_{Xi}(t)\eta_{X'j}(t')\rangle=\delta(t-t')\delta_{XX'}\delta_{ij}$, where $i$ and $j$ are cell indices, $X$ and $X'$ the modeled variables, $\delta_{XX'}$ and $\delta_{ij}$ are Kronecker deltas and $\delta(t-t')$ is the Dirac delta. Note that, as the standard chemical Langevin equation (*Gillespie, 2000*), *Equation 5* recovers the deterministic limit when the cell sizes go to infinity.

Due to the presence of stochasticity and the fact of having a target that is able to follow the dynamics of its upstream regulator, the threshold on the target $\Theta_T$ results in a soft threshold on the ATML1 variable (see *Figure 7*). A soft threshold means that there is a range of ATML1 values in which a cell being in 4C will be likely to prevent mitosis, and therefore, become giant. The higher the ATML1 value the cell has in this range, the more likely will for a cell to become giant.

Integration of the resulting Langevin equations with the Îto interpretation was performed by using a variation of the Heun algorithm (*Carrillo et al., 2003*) with an absorptive barrier at 0 to prevent negative values of the modeled variables. Growth and its dilution-derived effects were considered deterministic, and were integrated with an Euler algorithm. The integration time step was set to $dt = 0.1$. Note that stochasticity was also introduced in the initial conditions of the modeled variables and when resetting the timer variable after cell division (*Equation 4*). Cells divide according to a shortest path rule in which the new wall pass through the center of mass of the dividing cell (*Sahlin and Jönsson, 2010*). Daughter cells have the same initial ATML1 and Target concentrations at birth, but can have different sizes. After dividing, these cells will acquire different initial timer values due to the noise term in *Equation 4*. For the sake of simplicity, no mechanical interactions were implemented to the simulated tissue.

Unless otherwise stated, simulation parameters were set as described in *Table 1*. We set uniformly distributed random initial conditions for ATML1 and Target variables within the interval $[0,1]$ and $[0,0.1]$, respectively. Timer initial conditions were set in correlation to the cell size of the initial template, following the expression.

$$Timer_i(t=0) = \frac{0.8\,\theta_{C,D}}{E_{Max}-E_{Min}}(E_i(t=0)-E_{Min}) + 0.1\,\theta_{C,D}\big(1-U_i'\big)\,, \tag{6}$$

**Table 1.** Main parameter values used for simulations in *Figures 7* and *8* and *Figure 7—supplements 1–4*. We omit time and concentration units, since all are considered arbitrary.

| Parameter | Description | Values |
|---|---|---|
| $P_A$ | ATML1 basal production rate | 1.14 |
| $V_A$ | ATML1 auto-induction rate | 1.25 |
| $K_A$ | ATML1 concentration for half ATML1 auto-induction maximal rate | 1.9 |
| $n_A$ | Hill coefficient for ATML1 auto-induction | 5 |
| $G_A$ | ATML1 degradation rate | 1 |
| $V_T$ | Target maximal production rate | 10 |
| $K_T$ | ATML1 concentration for half ATML1-mediated target maximal production rate | 2 |
| $n_T$ | Hill coefficient for ATML1-mediated target induction | 1 |
| $G_T$ | Target degradation rate | 10 |
| $\Theta_T$ | Target threshold for inhibiting mitosis | 0.6 |
| $\Theta_{C,S}$ | Timer threshold for synthesis | 2 |
| $\Theta_{C,D}$ | Timer threshold for timer resetting | 3 |
| $P_C$ | Timer basal production rate | 0.1 |
| $E_0$ | Characteristic effective volume | 15 |
| | Exponential radial growth rate | 0.007 |
| | Exponential added growth rate to the vertical direction | 0.012 |

being $U'_i$ an uniformly distributed random number defined in the interval [0,0.1], and $E_{Min}$ and $E_{Max}$ the minimal and maximal areas of the cells at the start of the simulation. This made larger cells being initiated at more advanced stages of the cell cycle, and hence, being more likely to divide. Ploidies were initially set to either 2C or 4C, depending on whether the initial timer values set by *Equation 6* were lower or higher than the S-phase timer threshold $\Theta_{C,S}$.

We assigned different parameter values based on experimental evidence when available. Threshold values of the timer for the synthesis phase and division checkpoint ($\Theta_{C,S}$ and $\Theta_{C,D}$ respectively) were assigned so that we could recover 2C and 4C percentages of cells in *atml1–3* mutants (*Figure 2G*) in regions of the parameter space in which no giant cells were formed. Given the chosen timer threshold values and an arbitrary basal timer production rate, simulations were integrated throughout 105 arbitrary time units, so that cells could undergo around three cell cycles (*Roeder et al., 2010*). Simulations scanning the parameter space were performed by using logarithmic spaced values of the ATML1 basal production rate ($P_A$) and linearly spaced values of the ATML1 auto-induction production rate ($V_A$). Specifically, we performed simulations on 121 logarithmically spaced $P_A$ values between 0 and 2.2, and 11 linearly spaced $V_A$ values between 0 and 2.5. From these parameter scans, $P_A$ and $V_A$ parameters were chosen for the simulations shown in *Figure 7* and *Figure 7—figure supplement 4*. $P_A$ and $V_A$ parameter values for representing the wild-type sepal in *Figure 7* were chosen so that there was a unimodal distribution of ATML1 concentration with similar CVs to the experimental CVs, giving rise to the same number of giant cells found in developing sepals. In particular, we aimed to have sepals that developed a total of 30 giant cells with 8C and higher ploidy, with approximately 17 of those cells being 16C and higher ploidies (see *Figure 2H*). To ensure that the target approximately followed and mimicked the dynamics of ATML1, we simulated a target with a higher degradation rate than ATML1 itself. To grow the sepal in a realistic manner, we provided a certain degree of anisotropy on the tissue growth parameters, as previously reported experimentally (*Hervieux et al., 2016*).

The computational implementation of the model was performed through the open source C++ Organism package, (http://dev.thep.lu.se/organism/; *Bozorg et al., 2014*; *Jönsson et al., 2006*). Data analysis and plots from simulation output were performed with Python 2.7, the Matplotlib package (*Hunter, 2007*) and MATLAB. See *Source code 2* for further details on the implementation of the model and the analysis of the simulated data. The visualization of the simulated growing sepals was performed with Paraview software (http://www.paraview.org).

## ROC analysis and threshold determination of the simulated data

ROC analysis was also applied to the ATML1 concentration maxima across the different simulated lineages, by following a similar procedure as for the experimental data (see *Receiver operator characteristics (ROC) analysis* section and *Source code 2* for details). Classes were also defined based on their final identity; lineages having 2C ploidy at the end of the simulation were considered small cells, while lineages having 8C ploidy or higher were considered giant cells. Lineages remaining in 4C ploidy at the end of the simulated time course were excluded of the analysis, given their unknown final fate.

The soft ATML1 threshold $\Theta_A^*$ was determined by finding the threshold assigned to the optimal (maximized difference between TPR and FPR) operating point of the ROC curve. Specifically, we used 30 different random subsamples of the small cell population with as much cells as the pool of giant cells, so that the total cost of misclassification of positive and negative cases for the threshold determination would remain equivalent and similar to the experimental analysis. As a result, the computed soft threshold $\Theta_A^*$ was defined as the mean of the 30 different optimal thresholds found using random subsamples. This subsampling method, when applied to the target maxima throughout 4C time courses, could accurately predict the hard threshold of the target variable imposed in the simulations $\Theta_T$, which we denote by $\Theta_T^*$ (*Figure 7—figure supplement 3B,E*). We represented the predicted thresholds as a dashed red line within a red shaded red region. This red region shows the standard deviation of the 30 optimal thresholds computed in the subsampling method. Note that sometimes the shaded red region is too small to be seen (e.g. see $\Theta_A^*$ in *Figure 7C*).

## Acknowledgements

We thank Joseph Cammarata, David Ehrhardt, Lilan Hong, Bruno Martins, Om Patange, Wojciech Pawlowski, Dana Robinson, Mariana Wolfner, Mingyuan Zhu, and Zach Zussman-Dobbins for comments on the manuscript. We would like to thank Anna Marks for her assistance screening for functional mCitrine-ATML1 rescue lines. We would like to thank Richard Smith for helpful discussions about MorphoGraphX. We would like to thank Grégoire Malandain, and Christophe Godin for their help in using ALT. We thank Behruz Bozorg for providing a geometrical template for the tissue simulation and some code for the cell division algorithm. We would like the thank Jeff Long, Shinobu Takada, and Keiko Torii for providing transgenic lines. This work was supported by NSF IOS Plant, Fungal, and Microbial Developmental Mechanisms grants IOS-1256733 and IOS-1553030 (AKHR). This work was further supported by the Gatsby Charitable Foundation (GAT3272/GLC to JL and GAT3395/PR4 to HJ) and the Swedish Research Council (VR2013:4632 to HJ). JT and PFJ acknowledge postdoctoral fellowships provided by the Herchel Smith Foundation. This work made use of the Cornell Center for Materials Research Shared Facilities, which are supported through the NSF MRSEC program (DMR-1120296).

## Additional information

### Funding

| Funder | Grant reference number | Author |
| --- | --- | --- |
| National Science Foundation | IOS-1553030 | Adrienne H K Roeder |
| Gatsby Charitable Foundation | GAT3272/GLC | James C W Locke |
| Vetenskapsrådet | VR2013:4632 | Henrik Jönsson |
| Herchel Smith Foundation | | José Teles<br>Pau Formosa-Jordan |
| National Science Foundation | IOS-1256733 | Adrienne HK Roeder |
| Gatsby Charitable Foundation | GAT3395/PR4 | Henrik Jönsson |

The funders had no role in study design, data collection and interpretation, or the decision to submit the work for publication.

### Author contributions

HMM, Conceptualization, Data curation, Formal analysis, Investigation, Visualization, Methodology, Writing—original draft, Writing—review and editing, Biological experiments and imaging, Design of image quantification pipeline, Image analysis, Interpretation of results; JT, Software, Formal analysis, Methodology, Writing—original draft, Writing—review and editing, Design of image quantification pipeline, Development of image analysis code, Image analysis, Interpretation of results; PF-J, Software, Investigation, Writing—original draft, Writing—review and editing, Computational model development, Interpretation of results; YR, Software, Methodology, Design of image quantification pipeline; RS-B, Resources; Creating ATML1 over expression lines; GI, Resources, Creating ATML1 over expression lines; HJ, Software, Supervision, Funding acquisition, Project administration, Writing—review and editing, Computational model development, Interpretation of results; JCWL, Supervision, Funding acquisition, Project administration, Writing—review and editing, Computational model development, Interpretation of results; AHKR, Conceptualization, Data curation, Supervision, Funding acquisition, Investigation, Writing—original draft, Project administration, Writing—review and editing, Interpretation of results

### Author ORCIDs

Heather M Meyer, http://orcid.org/0000-0002-9991-6312
José Teles, http://orcid.org/0000-0003-1373-8250
Pau Formosa-Jordan, http://orcid.org/0000-0003-3005-597X
Yassin Refahi, http://orcid.org/0000-0001-6136-608X
Henrik Jönsson, http://orcid.org/0000-0003-2340-588X

James C W Locke, http://orcid.org/0000-0003-0670-1943
Adrienne H K Roeder, http://orcid.org/0000-0001-6685-2984

## Additional files

### Supplementary files

• Supplementary file 1. A zip file containing both Raw data and selected lineages for *pATML1::mCitrine-ATML1* (mCitrine-ATML1), *PDF1::GFP-ATML1* (PDF1), and *lgo-2;pATML1::mCitrine-ATML1* (lgo) flowers. See readme files within the different folders for further information. All raw image confocal tif files and example image processing files may be downloaded from: 10.7946/P29G6M

• Source code 1. MATLAB code for all image quantification and analysis, as well as receiver operator characteristics (ROC) analysis as described in the Materials and methods section.

• Source code 2. Code for simulating ATML1 dynamics in a growing tissue. Scripts for the analysis and representation of the simulation results are also provided. See readme files within the different folders for further information.

### Major datasets

The following dataset was generated:

| Author(s) | Year | Dataset title | Dataset URL | Database, license, and accessibility information |
|---|---|---|---|---|
| Meyer HM, Teles J, Formosa-Jordan P, Refai Y, San-Bento R, Ingram G, Jönsson H, Locke JCW, Roeder AHK | 2016 | Fluctuations of the transcription factor ATML1 generates the pattern of giant cells in the *Arabidopsis* sepal | http://dx.doi.org/10.7946/P29G6M | Publicly available at Cyverse (http://www.cyverse.org/) |

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
