## [Decision Letter]

Thank you for submitting your article "Fluctuations of the transcription factor ATML1 generate the pattern of giant cells in the *Arabidopsis* sepal" for consideration by *eLife*. Your article has been favorably evaluated by Christian Hardtke as the Senior Editor and four reviewers, one of whom is a member of our Board of Reviewing Editors. The following individual involved in review of your submission has agreed to reveal their identity: Steven Maere (Reviewer #4).

The reviewers have discussed the reviews with one another and the Reviewing Editor has drafted this decision to help you prepare a revised submission. Overall, the experimentalists thought the work was well done, but there were several additional controls and tests of the model that were needed to make this a truly solid story. The modeling was evaluated by another expert who again finds the work generally strong but offered some cautions in interpretations. All felt that the authors should be commended for the very clear writing and explanation of complex data in an accessible way.

Summary:

During development, patterns emerge in tissues and organs. The underlying basis for these patterns is a topic of broad interest in the community. Using the *Arabidopsis* sepal, an organ with a stereotyped size and shape and an epidermis consisting of several cell types, the authors use live cell imaging and modeling to investigate how the pattern of giant cells (GC) arises. The authors provide evidence that stochastic gene expression, a widespread phenomenon that arises from the nature of gene expression as it responds to intrinsic and extrinsic sources of variation, enables equivalent cells to adopt fates different from some of their neighbors, dependent upon cell-autonomous accumulation of a particular key regulator. In this manuscript the authors present a case for ATML1 being such a factor, whose activity in promoting GC fate is determined by stochastic levels that exceed a threshold.

Essential revisions:

1) Better test of the need for ATML1 in G2

High ATML1 levels are moderately predictive of GC identity (.74 or a.5-random to 1.0 absolutely predictive scale). This is OK, but it makes it all the more important to add in something that suggests the ATML1 level is a major component driving (and not simply reflecting) the fate.

Two possible ways to address this are: (1) The best way to do this would be to increase levels of ATML1 only during G2 and only during G1 and test whether these manipulations fit their model. I don't know all the tools available-in other systems there are G1, S and G2/M specific promoters and there are degradation elements linked to the cell cycle. Could these be employed? (2) If the first test is technically impossible, then another part of their model is that there is a feedback of ATML1 on its own expression. The ATML1 site from their ATML1 or PDF1 promoters could be removed to see whether this makes an important contribution.

2) Provide more convincing evidence that correlation of expression patterns (bursts in G2) aren't more general feature of TFs in these cells.

Reviewers were concerned by the use of *pSEC24A::H2B-GFP* abundance as a control for whether the relative broad variation in *pATML1::mCitrine-ATML1* abundance in the developing sepal is a general phenomenon. H2B is quite different from ATML1 in that it is incorporated in the nucleosomes and likely much more abundant compared to regular transcription factors. Also a different fluorescent probe is used. To prove that varying expression of ATML1 in the sepal epidermis is not a common feature among transcription factors, it would be better to use an unrelated mCitrine-TF fusion as control.

3) Some model interpretations are a bit overstretched and should be reconsidered.

For instance, the assertion in the fifth paragraph of the subsection “A model with stochastic fluctuations of ATML1 reproduces giant cell patterning”, that the model correctly recapitulates G2-mediated giant cell fate specification and that the lower AUC values recovered in 2C stages than in 4C in the model indicate 'that high ATML1 levels during the G2 phase of the cell cycle are important for giant cell fate commitment' is not really justified (or at least does not add anything to the experimental observations to this effect). Since a hard threshold for the 'Target' gene to start endoreduplication is hard-coded specifically in the G2 phase of the model, no other outcome could have been expected. Instead, I think the major contribution of the model is that it shows that stochasticity in ATML1 expression alone, associated with a hard threshold on ATML1 levels to induce giant cell formation, is not sufficient to explain the observations, and that it predicts instead that a hard G2-associated threshold on another stochastically influenced downstream factor, causing the ATML1 threshold to become soft, is needed to explain the imperfect relationship between high ATML1 expression in G2 and the induction of giant cell formation. It seems that additionally this downstream factor needs to have a higher degradation rate so that the 'Target' follows the dynamics of ATML1 (which makes sense, although the alternative was not tested), and that the auto-induction rate of ATML1 should not be too strong. These are in my view the real predictions of the model, the fact that e.g. overexpression of ATML1 in the model would lead to an ectopic giant cell phenotype can already be deduced from the model form without running simulations.

[Editors' note: further revisions were requested prior to acceptance, as described below.]

Thank you for resubmitting your work entitled "Fluctuations of the transcription factor ATML1 generate the pattern of giant cells in the *Arabidopsis* sepal" for further consideration at *eLife*. Your revised article has been favorably evaluated by Christian Hardtke as the Senior Editor, and a Reviewing Editor.

The manuscript has been significantly improved in many places, but there is still one issue to address.

Because you cannot supply ATML1 at a specific phase in the cell cycle, nor delete promoter elements, but instead rely on following endogenous ATML1 transcripts after induction to infer a feedback loop, this last experiment must be very carefully done and interpreted.

A concern is whether the technical set up (and point at which ATML1 is monitored) has the capacity to report a "strong" feedback and indeed, what level of transcriptional up-regulation would be considered strong feedback. How can the experiment be calibrated? You will either need to cite evidence from similar experiments (same induction system and timing and tissue for monitoring) that put the 1.7 fold increase in context, or do this type of control experiment yourselves.

---

## [Author Response]

*Essential revisions:*

*1) Better test of the need for ATML1 in G2*

*High ATML1 levels are moderately predictive of GC identity (.74 or a.5-random to 1.0 absolutely predictive scale). This is OK, but it makes it all the more important to add in something that suggests the ATML1 level is a major component driving (and not simply reflecting) the fate.*

*Two possible ways to address this are: (1) The best way to do this would be to increase levels of ATML1 only during G2 and only during G1 and test whether these manipulations fit their model. I don't know all the tools available-in other systems there are G1, S and G2/M specific promoters and there are degradation elements linked to the cell cycle. Could these be employed? (2) If the first test is technically impossible, then another part of their model is that there is a feedback of ATML1 on its own expression. The ATML1 site from their ATML1 or PDF1 promoters could be removed to see whether this makes an important contribution.*

We will address each of the three points raised here.

First, we agree that expressing ATML1 specifically during different stages of the cell cycle would be a great experiment; however, we also agree that it is technically impossible at this time. Overexpression of ATML1 in internal tissues is seedling lethal (Peterson et al., 2013; Takada et al., 2013). Since cell cycle genes are expressed in dividing cells in all cell layers at all stages of development, expressing ATML1 under one of these promoters would cause lethality. In addition, while there are well characterized combinations of promoters and degradation elements that can give rise to expression in S through early G2 and late G2 through M phase, there are currently no equivalently characterized promoters and degradation elements for G1 in *Arabidopsis* (Yin et al., 2014).

Second, we tested the feedback of ATML1 on its own expression using an ATML1 inducible system previously published (Peterson et al., 2013; Takada et al., 2013). We are doing this in place of the experiment proposed (mutating the ATML1 binding site in the ATML1 promoter) because while one ATML1 binding site has been characterized in the ATML1 promoter (Takada and Jürgens, 2007), there may be additional ATML1 binding sites within the promoter according to our searches with the ATML1 consensus binding sequences (Abe et al., 2001; Nakamura et al., 2006). This would make the deletion experiment difficult to interpret. Using the inducible system, we found that ATML1 works in a weak positive feedback loop as predicted by the model; endogenous ATML1 expression increased by 1.7 fold when ATML1 was induced with estradiol for 24 hours in inflorescences (Figure 7—figure supplement 4).

“To test the type of feedback of ATML1 on itself, we determined whether induction of ATML1 could rapidly stimulate transcription of the endogenous ATML1 gene using qPCR (Peterson et al., 2013). We found that ATML1 indeed acts in a weak positive feedback loop, increasing endogenous ATML1 expression by 1.7 fold within 24 hours (Figure 7—figure supplement 4). This result is consistent with our model, which suggests that ATML1 acts in a weak positive feedback loop.”

Third, and most critically, our genetics show that ATML1 is a major component driving giant cell formation not merely a reflection of giant cell fate. In Figure 2, we have shown that ATML1 overexpression is sufficient to produce ectopic giant cells, which turn on a giant cell fate marker (Figure 2). In contrast, overexpression of the cell cycle regulator LGO, which acts genetically downstream of ATML1, is insufficient to turn on the same giant cell marker (Roeder et al., 2012). Likewise, *atml1* mutants lose giant cells. To further validate this point, we now have added that inducing overexpression of ATML1 produces ectopic giant cells (Figure 2—figure supplement 1). Finally, we note that when we look at ATML1 peak expression specifically in the G2 phase of the cell cycle, we see that our predictability increases to 0.80, 0.80, 0.80, and 0.84.

“To validate that ATML1 alone is sufficient to drive giant cell formation, we induced *ATML1* expression in inflorescences using an ATML1 estradiol-inducible line. Ectopic giant cells formed on the sepal five days after being treated with 10µM estradiol (Figure 2—figure supplement 1). Overall, these results suggest that high levels of ATML1 are sufficient to induce sepal epidermal cells to adopt giant cell identity and can force a deterministic all-giant cell pattern.”

2) Provide more convincing evidence that correlation of expression patterns (bursts in G2) aren't more general feature of TFs in these cells.

*Reviewers were concerned by the use of pSEC24A::H2B-GFP abundance as a control for whether the relative broad variation in pATML1::mCitrine-ATML1 abundance in the developing sepal is a general phenomenon. H2B is quite different from ATML1 in that it is incorporated in the nucleosomes and likely much more abundant compared to regular transcription factors. Also a different fluorescent probe is used. To prove that varying expression of ATML1 in the sepal epidermis is not a common feature among transcription factors, it would be better to use an unrelated mCitrine-TF fusion as control.*

We have added VIP1-mCitrine and AP2-2XYpet controls. (Tian et al., 2004; Wollmann et al., 2010). VIP1 encodes a bZIP transcription factor. Although AP2 is not fused to mCitrine, we choose to include it because of its well-characterized role in floral organ identity. Both VIP1-mCitrine and AP2-2XYPet exhibit a lower CV than mCitrine-ATML1, confirming that not all transcription factors are expressed variably in developing sepals (Figure 3 and Figure 3—figure supplement 1).

“To see whether other genes also exhibit variable expression similarly to mCitrine-ATML1 in the developing sepal nuclei, we measured the expression of two fluorescently-tagged transcription factors, VIP1-mCitrine (pVIP1::VIP1-mCitrine) and AP2-2XYpet (pAP2::AP2-2XYpet), and the *SEC24A* transcriptional reporter (*pSEC24A::H2B-GFP*). […] We found that mCitrine-ATML1 concentrations were approximately twice as variable as the other reporters (Figure 3, Figure 3—figure supplement 1; VIP1 sepals show a mean CV of approximately 0.12; AP2 sepals show a mean CV of approximately 0.14; SEC24A sepals show a mean CV of approximately 0.12), suggesting that varying expression levels in sepal epidermal cells is not a common feature observed for every gene.”

3) Some model interpretations are a bit overstretched and should be reconsidered.

*For instance, the assertion in the fifth paragraph of the subsection “A model with stochastic fluctuations of ATML1 reproduces giant cell patterning”, that the model correctly recapitulates G2-mediated giant cell fate specification and that the lower AUC values recovered in 2C stages than in 4C in the model indicate 'that high ATML1 levels during the G2 phase of the cell cycle are important for giant cell fate commitment' is not really justified (or at least does not add anything to the experimental observations to this effect). Since a hard threshold for the 'Target' gene to start endoreduplication is hard-coded specifically in the G2 phase of the model, no other outcome could have been expected. Instead, I think the major contribution of the model is that it shows that stochasticity in ATML1 expression alone, associated with a hard threshold on ATML1 levels to induce giant cell formation, is not sufficient to explain the observations, and that it predicts instead that a hard G2-associated threshold on another stochastically influenced downstream factor, causing the ATML1 threshold to become soft, is needed to explain the imperfect relationship between high ATML1 expression in G2 and the induction of giant cell formation. It seems that additionally this downstream factor needs to have a higher degradation rate so that the 'Target' follows the dynamics of ATML1 (which makes sense, although the alternative was not tested), and that the auto-induction rate of ATML1 should not be too strong. These are in my view the real predictions of the model, the fact that e.g. overexpression of ATML1 in the model would lead to an ectopic giant cell phenotype can already be deduced from the model form without running simulations.*

We agree with the reviewer that some of the modeling claims of the paper should be made clearer. We have moved and adjusted the modeling section to appear at the end of the paper, which has made it clearer what is a prediction of the model. We have also provided model files (Source code 2) to allow readers to more easily simulate the model. Additionally, we have provided code files (Source Code 1) for the image analysis and quantification protocol.

In discussing the reviewers’ point that ‘the assertion in the fifth paragraph of the subsection “A model with stochastic fluctuations of ATML1 reproduces giant cell patterning”, that the model correctly recapitulates G2-mediated giant cell fate specification and that the lower AUC values recovered in 2C stages than in 4C in the model indicate 'that high ATML1 levels during the G2 phase of the cell cycle are important for giant cell fate commitment' is not really justified (or at least does not add anything to the experimental observations to this effect)’.

We have altered the original text:

“Consistent with our experimental observations, we found lower AUC values in 2C stages than in 4C, indicating that high ATML1 levels during the G2 phase of the cell cycle are important for giant cell fate commitment (Figure 5).”

To:

“Consistent with our experimental observations, we found lower AUC values in 2C stages than in 4C. This supports our hypothesis that high ATML1 levels during the G2 phase of the cell cycle are important for giant cell fate commitment (Figure 7 and Figure 7—figure supplement 3).”

We agree with the reviewer that a key prediction of the model is that the ATML1 feedback strength must not be too strong. We have stressed this point in the text, and have made it clear that our simulations where we vary the feedback strength in the ATML1 circuit predict that the system cannot contain a strong positive feedback loop. We now have further experimental evidence for this prediction (In answer to point 1), showing that the ATML1 autoregulatory feedback is weak (Figure 7—figure supplement 4), and mention this in the model Results section:

“To test the type of feedback of ATML1 on itself, we determined whether induction of ATML1 could rapidly stimulate transcription of the endogenous ATML1 gene using qPCR (Peterson et al., 2013). We found that ATML1 indeed acts in a weak positive feedback loop, increasing endogenous ATML1 expression by 1.7 fold within 24 hours (Figure 7—figure supplement 4). This result is consistent with our model, which suggests that ATML1 acts in a weak positive feedback loop.”

We also agree with the reviewer that it is of interest that the model suggests that a fast degradation of the target can be used to recapitulate the threshold selection behavior. However, we have not ruled out other mechanisms for enabling the target to follow the dynamics of ATML1. We changed the sentence in Methods:

“To ensure that the target approximately followed and mimicked the dynamics of ATML1, we proposed that the target would have a higher degradation rate than ATML1 itself.”

to

“To ensure that the target approximately followed and mimicked the dynamics of ATML1, we simulated a target with a higher degradation rate than ATML1 itself.”

Finally, although we agree with the reviewer that ‘the fact that e.g. overexpression of ATML1 in the model would lead to an ectopic giant cell phenotype can already be deduced from the model form without running simulations’, we think thatour simulations of ATML1 over expression do show another non-trivial result. Our simulations show that it is possible to fine-tune the number of giant cells based on the levels of ATML1 expression, showing the plausibility of the threshold based patterning mechanism.

[Editors' note: further revisions were requested prior to acceptance, as described below.]

*The manuscript has been significantly improved in many places, but there is still one issue to address.*

*Because you cannot supply ATML1 at a specific phase in the cell cycle, nor delete promoter elements, but instead rely on following endogenous ATML1 transcripts after induction to infer a feedback loop, this last experiment must be very carefully done and interpreted.*

*A concern is whether the technical set up (and point at which ATML1 is monitored) has the capacity to report a "strong" feedback and indeed, what level of transcriptional up-regulation would be considered strong feedback. How can the experiment be calibrated? You will either need to cite evidence from similar experiments (same induction system and timing and tissue for monitoring) that put the 1.7 fold increase in context, or do this type of control experiment yourselves.*

We thank the reviewers and editors for pointing out this concern about the induction experiment. To address it, we have completely repeated and replaced the induction experiment. We have changed the induction method to ensure *ATML1* was sufficiently induced in inflorescences and show that with 10 µM estradiol we have about a 700-fold induction of the transgene. Our induction is close to the 1000-fold induction that Takada et al. 2013 observed in seedlings with the same transgene after 7 days on 10µM estradiol plates. To put the induction in the context of our *ATML1* overexpression lines, we measured the total *ATML1* transcript after induction, including both induced and endogenous. 10µM estradiol produced a 7-fold induction of total *ATML1*, which is above the 5-fold increase we observed in the homozygous *ATML1* OX lines in which the sepals are covered in giant cells (Figure 2). These results are further consistent with the increased formation of giant cells we observed in these lines after induction (Figure 2—figure supplement 1).

In our inflorescences, the 10 µM estradiol induction results in a 1.5-fold induction in endogenous *ATML1* (Figure 7—figure supplement 4), again similar to the 1.7-fold induction of endogenous *ATML1* observed by Takada et al. in seedlings after 7 days. To put this fold change in context, we examined the induction of other genes downstream of ATML1 including *CER5* (2.3-fold), *FDH* (1.6-fold), *PDF2* (1.5-fold) and *PDF1* (1.2-fold) (Figure 7—figure supplement 4). Note that these downstream genes are induced with very similar fold changes as the endogenous *ATML1. CER5, FDH*, and *PDF1* do not encode transcription factors and therefore cannot act in a feedback loop. This suggests that the feedback of ATML1 on itself is not activating ATML1 further than other targets at the 48-hour time point. To further test the induction, we also induced with 0.1 µM or 1 µM estradiol and achieved intermediate levels of induction and activation of downstream genes.

The definition of ‘strong’ and ‘weak’ feedback in the paper come from our modeling simulations. By increasing the ratio between the basal and the ATML1-dependent production rates (*P_A_* and *V_A_* in the model), the model dynamics changes from having a single ‘low’ fixed-point (no to weak feedback), to a bistability region (strong feedback). We note that if the feedback strength was increased further you could be left with a single ‘high’ fixed point, but we do not consider this regime in the paper as this could not fit the experimental data. By definition the parameters chosen for the strong feedback cases in the simulations were in the bistability region (or at its boundary), leading to a long-tailed or bimodal distribution of ATML1 expression (Figure 7—figure supplement 4), which we do not observe experimentally (Figure 3). Our experimentally observed gradual increase in induced *ATML1* with increasing levels of estradiol further support the case for weak feedback in the system, as endogenous *ATML1* levels are not sensitive to small increases in exogenous *ATML1*. In the strong feedback case, sensitivity to *ATML1* induction increases as the system is bistable and easily reaches the high value state.

We have rewritten the manuscript to make these points clear, as shown below:

We modeled the different feedback strengths by varying the ratio between *ATML1* dependent and basal production rates, whilst keeping the number of predicted giant cells close to experimental values (Materials and methods).

To test the type of feedback of ATML1 on itself, we examined the effects of induction of *ATML1* on the transcription of the endogenous *ATML1* gene in inflorescences using qPCR (Peterson et al., 2013; Takada et al., 2013). We found that *ATML1* induction with 10 µM estradiol lead to total *ATML1* levels 7.1 times higher than the mock treated samples, and increased endogenous *ATML1* expression 1.5-fold within 48 hours (Figure 7—figure supplement 4). This level of induction was similar to that observed in other downstream genes, suggesting that the feedback of ATML1 on itself is not activating *ATML1* further than other targets at the 48-hour time point (Figure 7—figure supplement 4). The results are also consistent with a previous study carried out in seedlings after 7 days, where endogenous *ATML1* levels increased to 1.7-fold after induction (Takada et al., 2013).

To further test the properties of the feedback, we also induced with 0.1 µM or 1 µM estradiol and achieved intermediate levels of induction and activation of downstream genes. In our strong feedback simulations, the parameters chosen are on, or close to, the bistability region in the system, leading to a long-tailed or bimodal distribution of ATML1 expression (Figure 7—figure supplement 4), which we do not observe experimentally (Figure 3). Our experimentally observed gradual increase in induced *ATML1* with increasing levels of estradiol further supports the case for weak feedback in the system, as endogenous ATML1 levels are not sensitive to small increases in exogenous *ATML1*. In the strong feedback case, sensitivity to *ATML1* induction increases as the system is bistable and easily reaches the high value state. Thus, our results are consistent with weak feedback in the system.